# Sparse Autoencoders, Again?

Yin Lu [1]  Xuening Zhu [1]  Tong He [2]  David Wipf [2]

## Abstract

Is there really much more to say about sparse autoencoders (SAEs)? Autoencoders in general, and SAEs in particular, represent deep architectures that are capable of modeling low-dimensional latent structure in data. Such structure could reflect, among other things, correlation patterns in large language model activations, or complex natural image manifolds. And yet despite the wide-ranging applicability, there have been relatively few changes to SAEs beyond the original recipe from decades ago, namely, standard deep encoder/decoder layers trained with a classical/deterministic sparse regularizer applied within the latent space. One possible exception is the variational autoencoder (VAE), which adopts a stochastic encoder module capable of producing sparse representations when applied to manifold data. In this work we formalize underappreciated weaknesses with both canonical SAEs, as well as analogous VAEs applied to similar tasks, and propose a hybrid alternative model that circumvents these prior limitations. In terms of theoretical support, we prove that global minima of our proposed model recover certain forms of structured data spread across a union of manifolds. Meanwhile, empirical evaluations on synthetic and real-world datasets substantiate the efficacy of our approach in accurately estimating underlying manifold dimensions and producing sparser latent representations without compromising reconstruction error. In general, we are able to exceed the performance of equivalent-capacity SAEs and VAEs, as well as recent diffusion models where applicable, within domains such as images and language model activation patterns. A link to the code is here.

[1]School of Data Science, Fudan University [2]Amazon Web Services. Correspondence to: Yin Lu <yinlu23@m.fudan.edu.cn>, Xuening Zhu <xueningzhu@fudan.edu.cn>, Tong He <htong@amazon.com>, David Wipf <davidwipf@gmail.com>.

*Proceedings of the $42^{nd}$ International Conference on Machine Learning*, Vancouver, Canada. PMLR 267, 2025. Copyright 2025 by the author(s).

## 1. Introduction

Autoencoders represent a widely-applicable neural network model for obtaining useful low-dimensional representations of data, particularly when training labels are not available (Berahmand et al., 2024; Goodfellow et al., 2016). The basic architecture is simple: An encoder network first projects each observable data sample to a latent representation (usually low-dimensional), after which a decoder network attempts sample reconstruction based on this latent representation. Sparse autoencoders (SAE) (Ng, 2011; Ranzato et al., 2006; 2007) adopt an additional degree of design flexibility by accommodating a higher-dimensional latent space, but with the inclusion of an attendant regularization factor applied to the encoder output that encourages sparse representations (meaning most dimensions are pushed to zero). This capability facilitates *adaptive* latent representations reflecting complex underlying data structure, whereby the pattern of nonzeros can vary on a sample-by-sample basis unlike with a regular autoencoder.

Informally speaking, autoencoders (including SAEs) can be viewed as something like the horseshoe crabs of the deep learning kingdom, having existed for decades (or "eons") with more or less the same hardy conceptual design (Bourlard & Kamp, 1988; Hinton & Zemel, 1993; Le-Cun, 1987; Rumelhart et al., 1986). For example, the SAE models used for learning interpretable representations of present-day large language model (LLM) activation patterns (Bricken et al., 2023; Cunningham et al., 2024; Chaudhary & Geiger, 2024; Gao et al., 2024) are nearly identical in form to original architectures from long ago; likewise for a variety of other modern use cases (Lan et al., 2024; Movva et al., 2025; Pach et al., 2025; Stevens et al., 2025). One notable exception is the variational autoencoder (VAE) (Kingma & Welling, 2014; Rezende et al., 2014), which substitutes the deterministic latent space of traditional autoencoders, with an alternative stochastic representation. Although seemingly distinct from the SAE, it has recently been shown that VAEs can also address sparse autoencoding tasks in certain settings (Dai et al., 2021). Still, VAE models notwithstanding, given the long-lasting and durable usage of SAEs with minimal changes, it is natural to question the extent to which further innovation is possible on top of the original SAE design space, or if any such innovation is even needed at all.

We answer such questions in the affirmative based on the observation that both SAE and VAE designs have notable unresolved, yet distinct, shortcomings. While details will be deferred to Section 2, we summarize here that SAEs generally rely on multiple hyperparameters and potentially-nonconvex regularization factors that complicate both training and the interpretation of the latent representations produced by the encoder. In contrast, VAEs benefit from a hyperparameter-free and smoother loss surface, but are presently ill-equipped to produce adaptive sparse representations like the SAE. With this backdrop in mind, our contributions herein are encapsulated as follows:

- **Algorithmic:** Although both SAE and VAE models can reconstruct data using sparse encoder representations, we carefully detail their disjoint strengths and weaknesses in Section 2; subsequently we propose a hybrid VAE-like approach in Section 3 that delivers the best of both worlds. We refer to our model as *VAEase*, for an *easy*-to-apply *VAE* with *adaptive sparse estimation*.

- **Theoretical:** In Section 4 we rigorously prove that for data residing on a union of low-dimensional manifolds, globally-optimal VAEase solutions can provably recover the underlying manifold structure, adapting to multiple distinct manifolds, unlike existing VAEs. Likewise, we demonstrate simple illustrative settings whereby VAEase maintains fewer local minima relative to an analogous deterministic SAE model.

- **Empirical:** Finally, we present complementary experiments in Section 5 that show VAEase outperforms existing SAE and VAE architectures, as well as related diffusion models, across tasks such as estimating manifold dimensions or maximally sparse representations of large language model activations.

We conclude this section by emphasizing that, although our proposed approach explicitly relies on a variational chassis, the work itself as a whole is *not* directly related to generative modeling per se, the original purpose for which VAEs were designed. Instead, we are *repurposing* specific VAE attributes for new sparse autoencoding tasks.

## 2. Basics of Sparse Autoencoding

In this section we will first present a broad SAE formulation covering typical use cases. Later, we will demonstrate how VAEs can be leveraged for similar purposes, before concluding with a brief contextualization w.r.t. related diffusion models, as have recently been applied to learning manifold dimensions.

### 2.1. Canonical SAE Formulation

Suppose we collect $d$-dimensional data points $\mathbf{x} \in \mathcal{X} \subseteq \mathbb{R}^d$ drawn from some probability measure $\omega$ such that $\int_{\mathcal{X}} \omega(d\mathbf{x}) = 1$. Like traditional autoencoders (Bourlard & Kamp, 1988; LeCun, 1987; Rumelhart et al., 1986), an SAE model is composed of an *encoder* that maps samples $\mathbf{x}$ to a latent representation $\mathbf{z} \in \mathbb{R}^\kappa$, and a *decoder* that subsequently attempts to reconstruct $\mathbf{x}$ from $\mathbf{z}$. These modules are instantiated as functions given by $\boldsymbol{\mu}_z(\,\cdot\,;\phi) : \mathcal{X} \to \mathbb{R}^\kappa$ and $\boldsymbol{\mu}_x(\,\cdot\,;\theta) : \mathbb{R}^\kappa \to \mathbb{R}^d$ respectively, with the goal of learning parameters $\{\theta, \phi\}$ such that $\mathbf{x} \approx \boldsymbol{\mu}_x[\boldsymbol{\mu}_z(\mathbf{x};\phi);\theta]$ for all $\mathbf{x} \in \mathcal{X}$.[1]

What differentiates SAEs from regular auto-encoders (AEs) is that, in addition to seeking an encoder-decoder pairing capable of accurately reconstructing data samples, we also favor latent representations $\mathbf{z} = \boldsymbol{\mu}_z(\mathbf{x};\phi)$ that are *sparse* (Goodfellow et al., 2016; Ng, 2011; Ranzato et al., 2006; 2007). More formally, we ideally seek $\|\mathbf{z}\|_0 \ll \kappa$ (or at least approximately so), where $\|\cdot\|_0$ denotes the $\ell_0$-norm that counts the number of nonzero elements in a vector. To operationalize this preference, SAE models are generally trained with a loss in the form $\quad \mathcal{L}_{\text{SAE}}(\phi, \theta) :=$

$$\int_{\mathcal{X}} \left\{ \left\| \mathbf{x} - \boldsymbol{\mu}_x[\mathbf{z};\theta] \right\|_2^2 + \lambda_1 h(\mathbf{z}) \right\} \omega(d\mathbf{x}) + \lambda_2 \|\theta\|_2^2 \quad (1)$$

or a variant thereof, subject to $\mathbf{z} = \boldsymbol{\mu}_z(\mathbf{x};\phi)$. In this expression, $h : \mathbb{R}^\kappa \to \mathbb{R}$ serves as a penalty designed to approximate the $\ell_0$ norm while maintaining differentiability almost everywhere for training purposes. Candidates for $h$ include the $\ell_1$ norm $h(\mathbf{z}) = \sum_j |z_j|$ (the tightest convex relaxation to $\ell_0$ norm) or $h(\mathbf{z}) = \sum_j \log(|z_j| + \epsilon)$ where $\epsilon > 0$ is a small constant (Candes et al., 2008; Fazel et al., 2003); there exist many other approximation possibilities as well (Chen et al., 2017; Fan & Li, 2001; Palmer et al., 2006). However, a consequence of all these approximation candidates is the inevitable introduction of a scaling ambiguity, whereby elements of $\mathbf{z}$ can be pushed arbitrarily close to zero, while a targeted subset of $\theta$ can be proportionately increased towards infinity to compensate, such that the overall decoder reconstruction is unchanged. The penalty term $\|\theta\|_2^2$ in (1) is included to prevent this sort of degeneracy and ensure that $h$ induces non-trivial sparsity; see Appendix A.1 for further details. A similar effect can be accomplished by adding explicit constraints on the norm of $\theta$ during training (Cunningham et al., 2024) or allowing only the top $k$ values of $\mathbf{z}$ to be nonzero (Gao et al., 2024; Makhzani & Frey, 2013), although these approaches are effectively substituting one form of hyperparameter for another. And finally, we note that $\lambda_1$ and $\lambda_2$ in (1) are non-negative trade-off parameters balancing regularization effects and reconstruction error.

---

[1] We remark that the range of $\boldsymbol{\mu}_x$ need not be $\mathcal{X}$ exactly, and perfect reconstructions will not necessarily be feasible at all times.

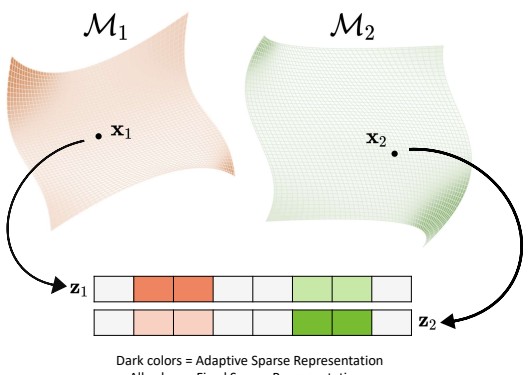

Dark colors = Adaptive Sparse Representation
All colors = Fixed Sparse Representation

Figure 1: *Fixed vs. adaptive sparsity.* Depicted are two manifolds $\{\mathcal{M}_1, \mathcal{M}_2\}$ with $\dim[\mathcal{M}_1] = \dim[\mathcal{M}_2] = 2$, and data points $\mathbf{x}_1 \in \mathcal{M}_1$ and $\mathbf{x}_2 \in \boldsymbol{M}_2$. *Adaptive* sparse representations require only two informative/active latent dimensions for either $\mathbf{x}_1$ or $\mathbf{x}_2$ (the dark colored elements of $\{\mathbf{z}_1, \mathbf{z}_2\}$ which vary across samples). Meanwhile, *fixed* sparse representations involve shared active dimensions across all samples (the union of dark and light elements).

**SAE Advantages:** After suitable training, the SAE formulation from (1) is quite powerful, capable of producing sparse representations of input samples drawn from $\omega$. In particular, a trained SAE can exhibit what is known as *adaptive* or *dynamic* sparsity (Ponti & Martins, 2024), whereby the support pattern (meaning location of nonzero elements) of a given latent code $\mathbf{z}$ can vary for different input samples $\mathbf{x}$. See Figure 1 for an illustration. This flexibility is critical because, as we will further detail in Section 4, it allows the SAE encoder to differentiate samples that lie on distinct manifold structures (as frequently encountered in practice (Brown et al., 2022)), or expand and contract the number of nonzero latent dimensions in accordance with sample-specific manifold complexity. As a representative example, the images of MNIST hand-written digits '1' may generally require fewer degrees-of-freedom to reconstruct than an observably more complex digits '8'.

**SAE Disadvantages:** On the downside, minimizers of (1) will typically be sensitive to the selection of $\lambda_1$ and $\lambda_2$, as well as the specific curvature of $h$. And for the latter, tighter approximations to the $\ell_0$ norm are relatively more difficult to optimize, with potentially numerous suboptimal local minimizers introduced in tandem with highly nonconvex penalties. Meanwhile, looser approximations such as the $\ell_1$ norm, although easier to minimize, may not generate optimally sparse latent representations. Hence for a given reconstruction error, $\|\mathbf{z}\|_0$ may be significantly larger compromising interpretability. Appendix A.3 provides an illustrative example of this contrast.

## 2.2. Alternative Variational Routes to SAE

Although originally proposed as a form of deep generative model for drawing new synthetic samples that approximately follow the ground-truth measure $\omega$, it has recently been shown that VAEs (Kingma & Welling, 2014; Rezende et al., 2014) can actually be repurposed to handle sparse autoencoding tasks (Dai et al., 2021). Broadly speaking, VAE models can be viewed as stochastic autoencoders, whereby the corresponding encoder and decoder modules now define *distributions* over $\mathbf{z}$ and sample reconstructions respectively. For models of continuous data (our focus), these distributions are commonly expressed as

$$
\begin{aligned}
q_\phi(\mathbf{z}|\mathbf{x}) &= \mathcal{N}\Big(\mathbf{z} \mid \boldsymbol{\mu}_z[\mathbf{x}; \phi], \mathrm{diag}\big[\boldsymbol{\sigma}_z(\mathbf{x}; \phi)\big]^2\Big) \\
p_\theta(\mathbf{x}|\mathbf{z}) &= \mathcal{N}\Big(\mathbf{x} \mid \boldsymbol{\mu}_x[\mathbf{z}; \theta], \gamma \boldsymbol{I}\Big),
\end{aligned}
\tag{2}
$$

where the functions $\boldsymbol{\mu}_z$ and $\boldsymbol{\mu}_x$ now serve as encode-decoder *mean* networks, while $\boldsymbol{\sigma}_z$ (upon squaring) defines a corresponding *variance* network. In accordance with typical use cases, the decoder variance is defined by a single parameter $\gamma > 0$, which may be trained along with other parameters. The corresponding VAE objective is given by $\mathcal{L}_{\text{VAE}}(\theta, \phi) :=$

$$
\int_{\mathcal{X}} \Big\{ -\mathbb{E}_{q_\phi(\mathbf{z}|\mathbf{x})}\big[\log p_\theta(\mathbf{x}|\mathbf{z})\big] + \mathbb{KL}\big[q_\phi(\mathbf{z}|\mathbf{x})\|p(\mathbf{z})\big]\Big\}\omega(dx)
\tag{3}
$$

with $p(\mathbf{z}) = \mathcal{N}(\mathbf{z}|\mathbf{0}, \boldsymbol{I})$; the first term above represents a reconstruction factor analogous to the first term in (1), while the latter KL term enacts VAE-specific regularization. Notably, for any encoder distribution $q_\phi(\mathbf{z}|\mathbf{x})$, it follows that $\mathcal{L}_{\text{VAE}}(\theta, \phi) \geq -\int_{\mathcal{X}} \log p_\theta(\mathbf{x})\omega(dx)$, with equality iff $q_\phi(\mathbf{z}|\mathbf{x}) = p_\theta(\mathbf{z}|\mathbf{x})$ almost surely. Using a simple reparameterization trick, it is possible to directly minimize (3) using SGD (Kingma & Welling, 2014; Rezende et al., 2014).

The stochastic VAE design obscures its ability to behave as a sparse autoencoder, and indeed, the mechanism for pruning unneeded dimensions of the latent space is distinct from canonical SAEs. While we defer details to prior work, the crux of the idea is that superfluous dimensions of $\mathbf{z}$ are flooded with uninformative white noise that is subsequently filtered by the decoder to avoid compromising accurate reconstructions. Mathematically, we have that $q_\theta(z_j|\mathbf{x}) \approx \mathcal{N}(z_j|0, 1)$ for unneeded dimension $j$, which in the sequel we will refer to as an *inactive* dimension. This is in contrast with a so-called *active* dimension $j'$ used to support reconstruction, which is characterized by $q_\theta(z_{j'}|\mathbf{x}) \approx \mathcal{N}\big(z_{j'}|\mu_z(\mathbf{x}; \phi)_{j'}, O(\gamma)\big)$. Because under relevant circumstances the optimal decoder variance will satisfy $\gamma \to 0$ during training, there exists a nearly deterministic pathway for producing arbitrarily accurate VAE reconstructions using these active dimensions (Dai et al., 2021; Zheng et al., 2022).

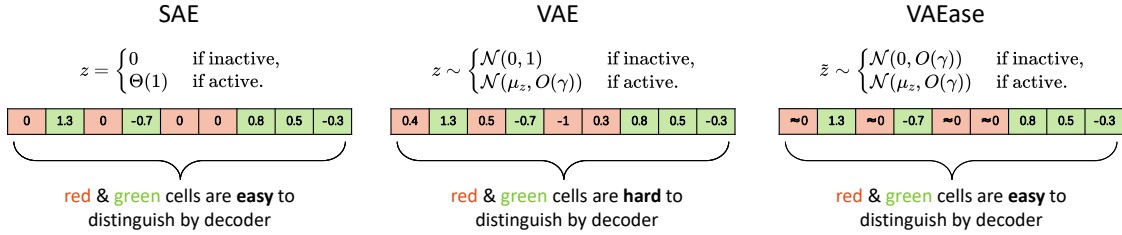

Figure 2: *Comparison among methods*. On the left SAE active (green) and inactive (red) dimensions are easily identifiable by the decoder, so accurate reconstructions are not disrupted when the locations of the nonzero elements *adaptively* shift from input sample-to-sample. Meanwhile in the middle figure, VAE active and inactive dimensions look similar for any given sample, which favors the VAE simply learning a *fixed* sparsity pattern such that the decoder knows which dimensions are active. And finally, on the right the VAEase-specific reparameterized representation $\widetilde{\mathbf{z}}$ mimics the original SAE behavior (as $\gamma \to 0$ near global optima), restoring the ability to adapt the sparsity pattern for each input sample. In this way, VAEase is more flexible in steering away from the fixed-sparsity representations favored by the vanilla VAE.

**VAE Advantages:** Since $\gamma$ can be learned along with other parameters as part of the overall probabilistic model, in principle sensitive hyperparameter tuning is not necessary. Additionally, if fixed sparsity is sufficient, then VAE models are potentially superior to SAEs because they provably have fewer suboptimal local minima in certain settings (Wipf, 2023). This is a desirable artifact of the stochastic VAE encoder, which is uniquely capable of *selectively smoothing away bad minima* while simultaneously preserving sharp global optimal with maximal sparsity.

**VAE Disadvantages:** The very stochastic smoothing that contributes to VAE advantages is also its Achilles heel. Specifically, whenever a given latent dimension $j$ is inactive for one or more data samples, this same dimension is more likely to be inactive for *all* data samples. This is because, unlike with SAE models, the first VAE decoder layer is generally forced to assign a zero-valued weight to incoming inactive dimensions, which are characterized by white noise that would otherwise contribute to high reconstruction errors at the decoder output. Please see Figure 2 for an illustrative example. And once a zero-valued weight is assigned during training, this dimension will remain permanently blocked across all input samples. Hence a de facto *fixed* sparsity paradigm is implicitly enforced (see Figure 1), in contradistinction to the more flexible adaptive/dynamic sparsity enjoyed by SAE models as discussed previously. This limitation is especially problematic in real-world data comprised of complex manifold structure.

### 2.3. Diffusion Models

We conclude Section 2 by briefly mentioning less direct connections between SAEs and diffusion models (DMs) (Ho et al., 2020; Sohl-Dickstein et al., 2015). As powerful generative architectures, DMs can be viewed as a hierarchical VAE, but with a parameter-free encoder that assumes $\kappa = d$

(Kingma et al., 2021; Luo). Because of this, DMs are not capable of directly producing sparse latent representations as SAEs and VAEs intrinsically do; they are therefore not a direct surrogate for sparse autoencoding tasks in general. However, it has nonetheless been recently demonstrated that using clever post-processing strategies, DMs are in fact capable of quantifying the dimensionality of data lying on low-dimensional manifolds (Horvat & Pfister, 2024; Kamkari et al., 2024; Stanczuk et al., 2024; Tempczyk et al., 2022). So in this sense their ancillary capabilities overlap with SAEs/VAEs as we will briefly explore in Section 5.3.

## 3. Sparse Autoencoding with (VA)Ease

In the previous section we highlighted contrasting strengths of SAE and VAE models: the former enjoys dynamic, input-adaptive sparsity but at the expense of additional hyperparameter and penalty tuning; meanwhile the latter requires no hyperparameters but can only reliably produce fixed sparsity patterns in practice. We now take steps towards achieving the best of both worlds via a novel, yet deceptively simple SAE/VAE hybrid loss formulation.

Our motivation comes from the fact that certain forms of *conditional* VAE models have in fact been shown to produce adaptive sparsity patterns (Zheng et al., 2022). CVAE models augment the decoder (and possibly the encoder and prior as well) to condition on some additional observable $\boldsymbol{y}$ leading to the revised distribution $p_\theta(\mathbf{x}|\mathbf{z}, \boldsymbol{y})$. From here, specially-designed decoder self-attention mechanisms can exploit $\boldsymbol{y}$ to selectively turn on and off latent dimensions following an input-specific manner, at least to the extent that $\boldsymbol{y}$ varies in accordance with $\mathbf{x}$.[2]

*But how to proceed more generally in normal circumstances*

---

[2]While it may be possible to achieve something similar without conditioning variables, the learning process is considerably more challenging. Please see further discussion in Appendix B.1.

*without access to helpful conditioning variables?* Our insight here is two-fold:

1. In VAE models, the $\boldsymbol{\sigma}_z$ network contains information about which dimensions are active, but it does not directly modulate reconstructions themselves as $\boldsymbol{\mu}_z$ does.

2. But we cannot just arbitrarily introduce $\boldsymbol{\sigma}_z$ to the decoder to achieve the adaptive sparsity we are after; this would directly lead to overfitting (see Appendix B.3). Instead, we would like to specifically leverage $\boldsymbol{\sigma}_z$ *only as a gating mechanism* to clamp the noise introduced by inactive dimensions. In this way the decoder weights themselves need not permanently close any particular inactive dimension across all inputs.

With these considerations in mind, we propose the following easily-applicable modification to a traditional VAE: We replace $p_\theta(\mathbf{x}|\mathbf{z})$ as defined in (2) with

$$p_\theta\big(\mathbf{x}|\mathbf{z}, \boldsymbol{\sigma}_z[\mathbf{x}; \phi]\big) = \mathcal{N}\Big(\mathbf{x} \mid \boldsymbol{\mu}_x[\widetilde{\mathbf{z}}; \theta], \gamma \boldsymbol{I}\Big),$$
$$\text{where} \quad \widetilde{\mathbf{z}} := \big(\mathbf{1} - \boldsymbol{\sigma}_z[\mathbf{x}; \phi]\big) \odot \mathbf{z} \qquad (4)$$

and $\odot$ denotes the Hadamard product. While seemingly a minor change, the consequences are profound in terms of how inactive dimensions are now handled. In particular, if $q_\theta(z_j|\mathbf{x}) \approx \mathcal{N}(z_j|0, 1)$ for inactive dimension $j$, then $\widetilde{z}_j \approx 0$, and so the decoder now receives an uninformative but still *deterministic* input that need not disrupt accurate reconstructions even for a decoder input layer with nonzero weights (analogous to the canonical SAE model). Meanwhile, for an active dimension $j'$ with $q_\theta(z_{j'}|\mathbf{x}) \approx \mathcal{N}(z_{j'}|\mu_z(\mathbf{x};\phi)_{j'}, O(\gamma))$, we have $\widetilde{z}_{j'} \approx \mu_z(\mathbf{x};\phi)_{j'}$ such that transmissible information for reconstructing $\mathbf{x}$ is not compromised. Figure 2 and Appendix B.3, convey the high-level picture of this phenomena.

We refer to the proposed VAE modification from above as *VAEase*, and summarize the newly-enabled capabilities in Table 1. Critically, VAEase preserves the original properties that make VAE models attractive for sparse autoencoding (i.e., no sensitive hyperparameters required, potential for local minima smoothing via a stochastic latent space), while removing the inflexible VAE favoritism towards fixed-sparsity solutions. Admittedly though, the motivations and claims thus far have been primarily supported by intuition at the expense of rigor; however, in the next section we place VAEase on a sound theoretical footing, with a much closer examination of properties relevant to sparse autoencoding.

## 4. Comparative Analysis

The notion of adaptive sparsity is particularly relevant when the underlying data distribution is spread across a union

Table 1: *Comparison of notable attributes.*

| | adaptive sparsity | local min smoothing | hyperparm free loss |
|---|---|---|---|
| SAE | ✓ | ✗ | ✗ |
| VAE | ✗ | ✓ | ✓ |
| VAEase | ✓ | ✓ | ✓ |

of low dimensional manifolds. Hence we choose this setting as the starting point of our analysis of VAEase. After formalizing concrete assumptions, we move on to examine the degree to which VAEase global minimizers align with ground truth manifold structure. We then conclude by describing a specific scenario whereby the VAEase objective has provably fewer local minima than its SAE counterpart.

### 4.1. Preliminaries

We now briefly lay out precursory definitions pertaining to our later data and modeling assumptions.

**Definition 4.1** (Data on a union of manifolds). Let $\mathcal{M} := \{\mathcal{M}_i\}_{i=1}^n$ denote a set of $n$ manifolds $\mathcal{M}_i \subset \mathbb{R}^d$, where $\dim[\mathcal{M}_i] = r_i \geq 0$ and there exists a diffeomorphic mapping $\psi_i : \mathcal{M}_i \to \mathbb{R}^{r_i}$. There is at most one manifold with $r_i = 0$. We then assume data $\mathbf{x} \in \mathcal{X} \subseteq \mathcal{M}$ with probability measure $\omega$ such that $\alpha_i := \int_{\mathcal{M}_i} \omega(d\mathbf{x}) > 0$ and $\sum_{i=1}^n \alpha_i = 1$. Finally, we stipulate that $\int_{\mathcal{S}_i} \omega(d\mathbf{x}) = 0$ for any sub-manifold $\mathcal{S}_i \subset \mathcal{M}_i$ satisfying $\dim[\mathcal{S}_i] < r_i$.

Brown et al. (2022) motivate the practical relevance of data following the spirit of Definition 4.1, along with the additional assumption that there is no overlap among manifolds. In this sense our definition is actually more flexible and the analyses which follow later in Section 4.2 hold with or without overlapping manifolds.[3] We next turn to a precise definition of the VAEase model class we seek to analyze. Given a resurgence of SAE use cases, including

**Definition 4.2** (Lipschitz VAEase objective). Assume that $\boldsymbol{\mu}_z$ and $\boldsymbol{\mu}_x$ are Lipschitz continuous functions of $\mathbf{x} \in \mathcal{X}$ and $\mathbf{z} \in \mathbb{R}^\kappa$ respectively. We then denote $\mathcal{L}_{\text{VAEase}}(\phi, \theta)$ as the resulting objective from (3), but with $p_\theta(\mathbf{x}|\mathbf{z})$ replaced by $p_\theta(\mathbf{x}|\mathbf{z}, \boldsymbol{\sigma}_z[\mathbf{x}; \phi])$ as defined in (4).

Lipschitz continuity assumptions for neural network models are quite common (Neyshabur et al., 2017; Bartlett et al., 2017) and are naturally achievable in practice by deep ReLU networks with bounded model weights. For our purposes,

---

[3]We are able to accommodate this greater flexibility because our results do not ultimately depend on the dimensionality of the tangent spaces at the boundaries of the support sets. Instead we rely on the optimization of integrals over the entire extent of each manifold such that individual manifold dimensions remain identifiable.

this assumption is merely included for mild technical reasons related to the adopted proof technique.

And finally, before proceeding to our main results we introduce two quantities (with origins stemming back to (Zheng et al., 2022)) that will collectively serve as a useful lens for differentiating VAE and VAEase capabilities.

**Definition 4.3** (Active dimension). For any fixed $\gamma$, let $\{\theta_\gamma, \phi_\gamma\}$ denote a minimizer of loss $\mathcal{L}_{\text{VAEase}}(\phi, \theta)$ given by Definition 4.2. We then specify that a latent dimension $j \in \{1, \ldots, \kappa\}$ is *active* at sample $\mathbf{x} \in \mathcal{X}$ if $\sigma_z^2(\mathbf{x}; \phi_\gamma)_j = O(\gamma)$ as $\gamma \to 0$. Additionally, we adopt $\mathcal{A}(\mathbf{x})$ to denote the set of active dimensions associated with $\mathbf{x}$.

The remaining so-called *inactive* dimensions will satisfy $\sigma_z^2(\mathbf{x}; \phi_\gamma)_j = 1 - O(\gamma)$; as shown in Appendix E.1, other possibilities will not occur almost surely.

**Definition 4.4** (Reconstruction error). Borrowing notation and definitions from above, we define the VAEase reconstruction error as

$$\mathcal{R} := \int_{\mathcal{X}} \mathbb{E}_{q_{\phi_\gamma}(\mathbf{z}|\mathbf{x})} \left[ \left\| \mathbf{x} - \boldsymbol{\mu}_x(\widetilde{\mathbf{z}}; \theta_\gamma) \right\|_2^2 \right] \omega(d\mathbf{x}). \quad (5)$$

We remark that equivalent *active dimension* and *reconstruction error* quantities can be defined for regular VAE models as well, the primary difference being $\mathcal{L}_{\text{VAE}}(\phi, \theta)$ replacing $\mathcal{L}_{\text{VAEase}}(\phi, \theta)$ as the optimization objective used for obtaining $\{\theta_\gamma, \phi_\gamma\}$.

### 4.2. Properties of Global Solutions

We now analyze properties of VAEase global minimizers that are particularly relevant to sparse autoencoding.

**Theorem 4.5.** *Assume data adhering to Definition 4.1 with $\sum_i r_i \leq \kappa$. Then as $\gamma \to 0$, global solutions to $\mathcal{L}_{\text{VAEase}}(\phi, \theta)$ achieve $\mathcal{R} = o(1)$ and*

$$\int_{\mathcal{M}_i} \mathbb{I}(|\mathcal{A}(\mathbf{x})| = r_i) \omega(d\mathbf{x}) = \alpha_i + o(1) \; \forall i. \quad (6)$$

Note that because $\alpha_i = \int_{\mathcal{M}_i} \omega(d\mathbf{x})$ and $\sum_i \alpha_i = 1$, (6) indicates that within each data manifold $\mathcal{M}_i$, the VAEase is almost surely using a number of active dimensions that matches $\dim[\mathcal{M}_i]$. Overall then, this result establishes fundamental capabilities of the VAEase model: *It can achieve arbitrarily good reconstruction error, all while relying on ideal active dimension counts that perfectly align with each constitute data manifold.* Meanwhile, a corresponding VAE cannot achieve the equivalent of Theorem 4.5.

**Corollary 4.6.** *Consider the VAE loss $\mathcal{L}_{\text{VAE}}(\phi, \theta)$ defined as in (3) using arbitrary Lipschitz continuous encoder/decoder networks (i.e., equivalent to those adopted by the VAEase). Then there exist datasets adhering to Definition 4.1 such that*

*global minimizers of $\mathcal{L}_{\text{VAE}}(\phi, \theta)$ in the limit $\gamma \to 0$ satisfy*

$$\int_{\mathcal{M}_i} \mathbb{I}(|\mathcal{A}(\mathbf{x})| \neq r_i) \omega(d\mathbf{x}) = \Theta(1) \quad (7)$$

*for one or more constituent manifolds $\mathcal{M}_i$.*

Critically, this result indicates that VAE models are *not* always capable of adapting the sparsity of their latent representations to ground-truth structure involving multiple manifolds. Instead, for reasons detailed in Appendix B.1, VAE models tend to favor solutions whereby

$$\int_{\mathcal{M}} \mathbb{I}(|\mathcal{A}(\mathbf{x})| = r) \omega(d\mathbf{x}) \approx 1, \quad (8)$$

assuming $r := \sum_i r_i \leq \min(\kappa, d)$ (we prove this claim for linear special cases in Appendix E.5). As such the VAE can readily estimate the *aggregated* manifold dimension, generally with a fixed set of active dimensions, but not necessarily that of the finer-grained *individual* manifolds themselves as our proposed VAEase model can. Complementary experiments in Section 5 will confirm this performance disparity.

### 4.3. Local Minima Smoothing

Unlike the VAE, with an appropriate selection of $h$ and $\{\lambda_1, \lambda_2\}$, the SAE model from (1) can in principle mimic the behavior of the VAEase global optima explored in Section 4.2. As a deterministic counterpart, this involves producing adaptive sparse solutions with negligible reconstruction error and a minimal number of nonzero latent dimensions aligned with each manifold. For example, we can accomplish this goal with $h(\mathbf{z}) = \|\mathbf{z}\|_0$ and suitable choices for $\{\lambda_1, \lambda_2\}$ (see Appendix A.2). But SAE adaptive sparsity comes at a cost: In addition to the need for tuning $\{\lambda_1, \lambda_2\}$ and choosing some workable/smooth approximation to the $\ell_0$ norm that allows for SGD training, the SAE loss potentially has a much larger constellation of sub-optimal local minimizers. While difficult to explicitly quantify under broad conditions, in this section we introduce a simplified regime whereby an explicit, exponential gap in local minima counts can be established.

**Theorem 4.7.** *Assume $\kappa = d$ and decoder $\boldsymbol{\mu}_x(\widetilde{\mathbf{z}}; \theta) = \mathbf{U} \, \text{diag}[\mathbf{w}] \widetilde{\mathbf{z}}$, where $\mathbf{U} \in \mathbb{R}^{d \times d}$ is a fixed/known orthonormal matrix and $\mathbf{w} \in \mathbb{R}^d$ are learnable parameters. Moreover, assume arbitrary encoder functions $\boldsymbol{\mu}_z$ and $\boldsymbol{\sigma}_z$. Then the VAEase loss defined w.r.t. an arbitrary input data point $\mathbf{x}$ will always have a unique minimum (local or global).*[4]

We now contrast with an analogous SAE model, assuming $h \in \mathcal{H}$, where $\mathcal{H}$ denotes the set of functions that are

---

[4]Technically speaking, there are actually multiple equivalent global minima because the overall loss is invariant to randomly flipping the signs of $\mathbf{z}$ and $\mathbf{w}$. But this inconsequential ambiguity is shared by all SAE and VAE models alike.

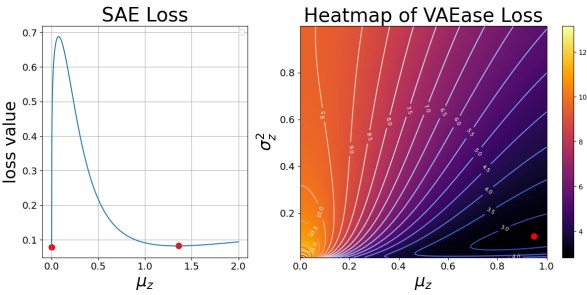

Figure 3: *Local minima smoothing example.* On the left a 1D slice of the SAE loss (details in Appendix D.1) has two distinct minima appear (red dots). Meanwhile, on the right VAEase under identical conditions has a single (global) minimum because of the latent smoothing afforded by $\sigma_z$.

symmetric about zero, concave and non-decreasing in $|z|$. Commonly-used sparsity-promoting penalties typically fall into this category (Chen et al., 2017), which includes the $\ell_1$ and $\ell_0$ norms, log-based penalties, and many other choices.

**Corollary 4.8.** *Under the same conditions as Theorem 4.7, for any $\{\lambda_1, \lambda_2\} \in \mathbb{R}_+^2$, there exists data samples $\mathbf{x}$ such that the SAE loss from (1) has $2^d$ distinct local minimizers.*[5]

Please see Figure 3 for a 1D visualization of the contrast between Theorem 4.7 and Corollary 4.8. Collectively, these results suggest that the SAE loss is likely more prone to sub-optimal local minima than VAEase. The latter uniquely benefits from a selective smoothing effect introduced by the stochastic latent space, whereby desirable sparse global solutions can be preserved (Theorem 4.5) while at least some distracting local minimizers are smoothed away (Theorem 4.7), ultimately because of the expectation over $q_\theta(\mathbf{z}|\mathbf{x})$. This phenomenon has been previously explored within the narrow context of fixed-sparsity VAE models (Wipf, 2023), but never generalized to the much more challenging adaptive sparsity scenario as we have done.

## 5. Empirical Validation

In this section we empirically compare VAEase against salient baselines on both synthetic and real-world datasets. Full experimental details, including data generation processes and model settings, are deferred to Appendix D.

### 5.1. Experiments with Known Ground-Truth

**Dataset generation.** Synthetic datasets are ideal in the sense that we have access to ground-truth manifold parame-

ters for explicitly quantifying model representations. To this end, we extend setups inspired from Zheng et al. (2022); Stanczuk et al. (2024) to generate two datasets composed of multiple manifolds. The first involves 500,000 points randomly assigned to $n = 3$ linear subspaces of 4 dimensions each embedded an ambient space with $d = 40$. The second more challenging case involves $n = 4$ nonlinear manifolds with dimensions $\{5, 5, 10, 10\}$ embedded in $d = 100$ dimensional space; 200,000 samples are generated per manifold by passing random noise through different MLP networks.

**Baseline models.** To compare against VAEase, we train an equivalent VAE baseline as well as three SAE variants differentiated by the penalty function adopted for $h$ in (1). Following prior work, we select SAE-$\ell_1$ for the standard $\ell_1$ norm as used by Cunningham et al. (2024), SAE-log whereby $h(z) = \log(|z| + \epsilon)$ ( $\epsilon > 0$ is a small constant) as motivated in Candes et al. (2008), and SAE-$T_k$ for the strategy of simply choosing the top/largest $k$ elements of $\mathbf{z}$ (per data sample) as advocated by Makhzani & Frey (2013); Gao et al. (2024). For the linear subspace dataset we adopt a linear decoder for all models (as is also used in recent application to modeling the intermediate activation layers of LLMs (Chaudhary & Geiger, 2024)); for the more complex nonlinear dataset we instead stack multiple MLP decoder layers. Note that for both datasets we set $k$, as required by SAE-$T_k$, to the size of the largest individual manifold.

**Evaluations.** After training, we evaluate models as follows. For each manifold, we sample new test points not seen during training and feed them into model encoders. We then use the average $\boldsymbol{\sigma}_z$ (for VAE models) or average absolute $\boldsymbol{\mu}_z$ (for SAE models), with the averages taken *within* each manifold, to estimate the active dimensions required. Ideally these averages should correspond with the ground-truth manifold dimension. For all experiments throughout this paper, we separate active from inactive dimensions using a simple, intuitive variance criteria: A partitioning threshold is chosen such that the average variance above (active dims) and below the threshold (inactive dims) is minimum. This concept, adapted from Xia et al. (2015), avoids dependency on subjective thresholds that vary from experiment to experiment. Results are displayed in Table 2, where VAEase clearly outperforms the other baselines. Note that all models produced negligible reconstruction errors ($\sim 10^{-3}$ or less). We also remark that SAE-$T_k$, even when $k = 4$ on the linear dataset, does *not* correctly estimate individual manifold dimensions; this is because the model fails to find a *consistent* set of active dimensions within each manifold.

### 5.2. Real-World Data

**Image data.** As we turn to real-world datasets, it is no longer feasible to expect near-zero reconstruction errors

---

[5]If an arbitrary loss $\mathcal{L}(u)$ is non-increasing in $u \in \mathbb{R}$ for all $u > u'$ for some constant $u'$, we also treat $u = +\infty$ as a local minima as there is no trajectory leading back to another local minimum without increasing the loss. But this inconsequential caveat only applies when $h$ is bounded from above.

Table 2: *Synthetic dataset results.* $AD_{\#}$ denotes active dims. for $\mathcal{M}_{\#}$. Closest result to ground-truth (GT) in **bold**.

| | Linear Subspaces | | | Nonlinear Manifolds | | | |
|---|---|---|---|---|---|---|---|
| | $AD_1$ | $AD_2$ | $AD_3$ | $AD_1$ | $AD_2$ | $AD_3$ | $AD_4$ |
| GT | 4 | 4 | 4 | 5 | 5 | 10 | 10 |
| SAE-$\ell_1$ | 10 | 11 | 8 | 20 | 26 | 39 | 29 |
| SAE-log | 8 | 8 | 8 | 15 | 28 | 39 | 31 |
| SAE-$T_k$ | 10 | 10 | 7 | 10 | 12 | 18 | 19 |
| VAE | 13 | 13 | 13 | 17 | 21 | 20 | 19 |
| VAEase | **5** | **4** | **4** | **5** | **5** | **11** | **11** |

or known ground-truth manifold structure. In this underdetermined regime, achieving a target reconstruction error using the fewest active dimensions is the core objective. To this end, we tune SAE hyperparameters such that the reconstruction error roughly matches VAE counterparts and then examine the required number of active dimensions. We apply this approach to MNIST (Deng, 2012) and Fashion-MNIST (Xiao et al., 2017) image datasets, both of which are likely to have rich underlying manifold structure. To better equip models for handling image data, we use ReLU convolutional layers for all encoder and decoder networks. Results are shown in Table 3, where we now report the number of active dimensions averaged across all test samples ($\overline{AD}$) as well as the reconstruction error (RE) for reference. VAEase achieves strong performance on average as the only method to rank within the top two for all categories. The closest competitor is SAE-$T_k$; however this approach relies explicitly on access to a data-dependent estimate of the manifold dimension applied through manual selection of $k$. In contrast, VAEase operates effectively without any such knowledge, a capability that is particularly valuable when obtaining such estimates are costly, or when there is wide variance across data samples.

To further explore VAEase attributes, Figure 4 provides a more fine-grained visualization of FashionMNIST results. From the displayed curves we observe that VAEase maintains a stable reconstruction error as more and more latent dimensions are sequentially masked out. Results on other datasets (not shown) are similar. As an additional side point, Appendix C demonstrates how VAEase active dimensions correlate with MNIST/Fashion image labels.

**Language model activation data.** We further explore orthogonal comparisons using a popular large language model (LLM) related use case. Specifically, SAE models have recently been applied to learning sparse reconstructions of the activation patterns produced within intermediate LLM layers for interpretability reasons (Bricken et al., 2023; Cunningham et al., 2024; Chaudhary & Geiger, 2024). In these scenarios there is no explicit ground-truth; however, as before the objective is to minimize the active dimensions sub-

ject to a nominal reconstruction error, the idea being that fewer dimensions are inherently more explainable.[6] To this end, using code from (Cunningham et al., 2024) we pass the Pile-10k dataset (Gao et al., 2020) through a Transformer model from the Pythia Scaling Suite (Biderman et al., 2023) and extract 2,098,176 samples from an intermediate activation layer, each with a dimensionality of $d = 512$. And since the dataset is drawn from highly diverse domains including literature, academic papers, and web text, it is reasonable to assume complex underlying data structure. We then trained VAEase and baseline models on these data and recorded results in Table 3, where we observe that VAEase simultaneously achieves the lowest RE and $\overline{AD}$ values. We also note that the VAE performance is particularly poor w.r.t. $\overline{AD}$. And yet in Appendix B.2 we show that even adding an arbitrarily complex self-attention layer to the otherwise linear decoder cannot fix the core VAE difficulty of identifying high-sparsity codes for LLM activations.

**Text embedding data.** As a final complementary application domain for evaluating VAEase, we consider text embedding data as formed at the output layer of certain LLM architectures. We choose such data because recent work has shown that sparse representations thereof (as obtainable from SAEs) encode rich semantic and causal relationships that are readily applicable to various forms of hypothesis generation (Movva et al., 2025) as well as interpretability studies (O'Neill et al., 2024). Following (Movva et al., 2025), we generate text embeddings for the Yelp dataset using the ModernBERT Embed model (Nussbaum et al., 2024) resulting in 200,000 samples with dimensionality $d = 768$. We then compare the sparse representations learned from VAEase training against baseline models (SAE-based and VAE) with results shown in Table 3. Similar to the LLM intermediate activation data from before, VAEase achieves significantly lower values for both $\overline{AD}$ and RE as desired. Meanwhile the VAE exhibits the poorest performance, likely attributable to uniformity in active dimensions across samples. We also remark that SAE-$T_k$ did *not* yield a smaller RE even when afforded a higher $\overline{AD}$ for all LLM/text-related experiments. We hypothesize that fixing the number of active dimensions ($k$=30) inherently constrains the model's capacity to adequately adapt across samples of varying complexity, hindering RE on average.

### 5.3. Comparison with Diffusion

As mentioned in Section 2.3, through suitable postprocessing DMs are in principle capable of estimating man-

---

[6]Pursuing LLM explanations is well outside the scope of our work. Instead, we are merely adopting this use case to showcase differences between VAEase and existing approaches on complex, high-dimensional data. Hence we focus on comparing active dimension counts, not downstream interpretations thereof.

Table 3: *Real-world dataset results.* RE = reconstruction error; $\overline{\text{AD}}$ = average # active dimensions in test set. Note that SAE-T$_k$ is given a dataset-dependent estimate of the manifold dimension through $k$, while VAEase *requires no such access to prior knowledge* and yet still maintains the best overall performance. Top result for each case in **bold**; second best is underlined.

|  | MNIST | | Fashion | | LLM-Act | | Text-Emb | |
|---|---|---|---|---|---|---|---|---|
|  | RE | $\overline{\text{AD}}$ | RE | $\overline{\text{AD}}$ | RE | $\overline{\text{AD}}$ | RE | $\overline{\text{AD}}$ |
| SAE-$\ell_1$ | 10.4 | 19.1 | 8.69 | 19.0 | 46.2 | 38.2 | 0.224 | 61.9 |
| SAE-log | 9.84 | 18.5 | 8.95 | 17.8 | 45.1 | 58.2 | 0.230 | 56.0 |
| SAE-T$_k$ | 10.1 | **16.0** | **8.11** | **15.0** | 45.1 | 30.0 | 0.217 | 30.0 |
| VAE | 9.81 | 18.0 | 8.42 | 18.1 | 48.0 | 87.9 | 0.277 | 90.0 |
| VAEase | **9.71** | 16.2 | 8.39 | **12.4** | **39.5** | **22.5** | **0.187** | **16.7** |

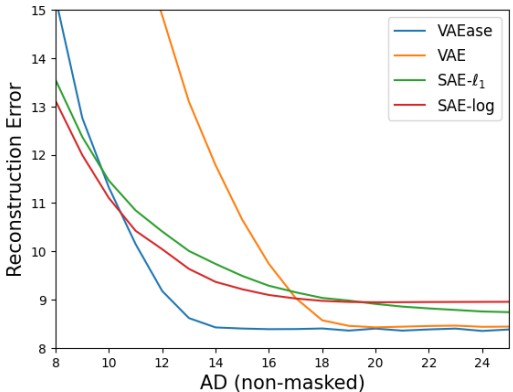

Figure 4: Reconstruction error (RE) curve as latent dimensions are sequentially masked out on FashionMNIST data. VAEase maintains a stable/flat RE with the fewest (non-masked) active dimensions, followed by a sharper rise once key informative dimensions start being removed. Note that SAE-T$_k$ is not included as its active dimensions are effectively fixed by the selection of $k$.

ifold dimensions (although not a corresponding mapping onto low-dimensional representations as SAE/VAE-based models do). Ideally, we would like to compare VAEase with diffusion in a real-world context, but we cannot rely on joint RE/$\overline{\text{AD}}$ evaluations as before, since DMs produce no analogous encode-decoder bottleneck reconstruction. Moreover, with real-world data we do not generally have access to ground-truth manifold dimensions that would otherwise facilitate head-to-head comparisons. In fact, even on MNIST there is considerable ambiguity, with wide-ranging estimates from 10 to over 100 (Pope et al., 2021; Tempczyk et al., 2022; Horvat & Pfister, 2024; Stanczuk et al., 2024). So instead we develop the following setup that is novel in the context of manifold dimension estimation.

First, we train a GAN (Goodfellow et al., 2014) with 16-dimensional input noise on MNIST data. Once trained, we then generate a *pseudo*-MNIST dataset by sampling this

Table 4: *Pseudo-MNIST comparison with diffusion models.*

| GT | DM (NB) | DM (FLIPD) | VAEase ($\overline{\text{AD}}$) |
|---|---|---|---|
| 16 | 105.94 | 169.81 | **14.93** |

model. By construction, these new images *will necessarily lie on a manifold with no more than* 16 *dimensions*, i.e., we now have access to a ground-truth upper bound. From here, we train VAEase and a DM on 500,000 samples and compare their respective estimates. For DM we used two recently-proposed post-processing techniques to estimate the manifold dimension: NB (Stanczuk et al., 2024) and FLIPD (Kamkari et al., 2024). Results are shown in Table 4, where VAEase provides a far more accurate estimate, being much closer to any possible value in the feasible range between 1 and 16 than either diffusion approach. We also remark that, because the pseudo-MNIST samples are visually quite similar to the originals (see Appendix D.4), it would appear that $r = 16$ (or possibly somewhat smaller) is actually a reasonable estimate for the true MNIST manifold dimension. This lends additional credence to the VAEase estimate of 16.2 from Table 3 (the same is *not* true for SAE-T$_k$ since the model was explicitly provided with $k = 16$).

# 6. Conclusions

Despite a lengthy and stable existence, we have argued that there is indeed still justification for revisiting the original design space of sparse autoencoders, again. In doing so we proposed a simple alternative architecture called VAEase, based on a novel retooling of the VAE encoder variance as an adaptive sparsity selector. This refinement allows an otherwise rigid, fixed-sparsity VAE latent space, to vary support patterns across input samples like a canonical SAE, even while retaining a hyperparameter-free energy along with stochastic smoothing of locally-minimizing solutions. And no additional model parameters, beyond those of a vanilla VAE, are required for implementing VAEase. Given the contemporary resurgence of creative SAE use cases, particularly as related to understanding properties vision and language models (Lan et al., 2024; Pach et al., 2025; Stevens et al., 2025) and beyond (Movva et al., 2025), we anticipate that VAEase may have widespread applicability moving forward.

## Impact Statement

This paper presents work designed to better understand and enhance the field of sparse autoencoders. While in principle machine learning models of this sort could be used for nefarious purposes, there is nothing specific to our work that stands out as a notable risk factor. And generally speaking, we would argue that by increasing transparency on model behaviors, we at least reduce the risk in inadvertent misuse or misapplication.

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

# A. Further Analysis of Sparse Autoencoding (SAE) Models

In this section we further elaborate relevant SAE modeling details that were omitted from the main text for space considerations. Later in Appendix B we provide complementary information regarding VAE models.

## A.1. Degeneracies in Deterministic SAE Models

We will now introduce a simplified scenario whereby minima of the SAE loss from (1) can have degenerate minima that undermine the central purpose of sparse autoencoding. Consider the case where the SAE decoder network is given by the linear function $\boldsymbol{\mu}_x[\mathbf{z}; \theta] = \mathbf{W}\mathbf{z}$, with $\theta = \mathbf{W} \in \mathbb{R}^{d \times \kappa}$. Furthermore, for some input sample $\mathbf{x}$, define

$$\mathbf{z}^* := \arg\min_{\mathbf{z}} \|\mathbf{x} - \mathbf{W}\mathbf{z}\|_2^2. \tag{9}$$

In general $\mathbf{z}^*$ will not have many or most values equal to or nearly zero; however, we can always fabricate such a solution by a trivial transformation. Specifically, we can make all elements of $\mathbf{z}^*$ arbitrarily small by the rescaling $\mathbf{z}^* \to \alpha\mathbf{z}^* \approx 0$, while maintaining an equivalent reconstruction error of $\mathbf{x}$ via the revised weights $\mathbf{W} \to \alpha^{-1}\mathbf{W}$. In this way we can reduce any smooth penalty factor $h(\mathbf{z})$, with minimum at zero, to an arbitrarily small value just by rescaling. But of course this defeats the whole purpose of including a sparse penalty in the first place. It is precisely because of this possibility that we must introduce $\|\theta\|_2^2$ into (1); by including this factor we limit the compensatory growth of $\theta = \mathbf{W}$ such that $h$ can produce non-trivial sparse solutions as desired.

While this particular scenario with a linear decoder is chosen for transparency, it nonetheless illustrates a broader complication with this type of model. Even if we were to deepen the decoder network by stacking additional nonlinear layers, the same scaling ambiguity will generally exist unless some form of restriction on $\theta$ is included.

One exception to this rule is when $h$ is replaced by a top-$k$ constraint (Gao et al., 2024; Makhzani & Frey, 2013) as mentioned in Section 2.1. Although this substitution mitigates scaling degeneracies, it introduces new complications. First, the top-$k$ constraint produces discontinuities in the loss surface that may interfere with timely convergence of the learning process. But secondly, and perhaps more importantly, it forces the latent representation of every input $\mathbf{x}$ to have exactly the same number of degrees-of-freedom (i.e., $k$), regardless of varying levels of input complexity.

## A.2. Maximal Sparsity Using an Infeasible SAE Model

As a point of reference, it is worthwhile to consider an SAE model capable of idealized adaptive sparsity akin to VAEase. To this end, consider the SAE loss from (1) actualized with $h(\mathbf{z}) = \|\mathbf{z}\|_0$ and $\lambda_2 = 0$. In this case we have

$$\mathcal{L}_{\text{SAE}}(\phi, \theta) = \int_{\mathcal{X}} \left\{ \left\| \mathbf{x} - \boldsymbol{\mu}_x[\mathbf{z}; \theta] \right\|_2^2 + \lambda_1 \|\mathbf{z}\|_0 \right\} \omega(d\mathbf{x}) \equiv \int_{\mathcal{X}} \left\{ \tfrac{1}{\lambda_1} \left\| \mathbf{x} - \boldsymbol{\mu}_x[\mathbf{z}; \theta] \right\|_2^2 + \|\mathbf{z}\|_0 \right\} \omega(d\mathbf{x}) \tag{10}$$

subject to $\mathbf{z} = \boldsymbol{\mu}_z(\mathbf{x}; \phi)$. In the limit as $\lambda_1$ becomes small, minimizing $\mathcal{L}_{\text{SAE}}(\phi, \theta)$ in this form is asymptotically equivalent to solving the constrained problem

$$\min_{\phi, \theta} \int_{\mathcal{X}} \|\mathbf{z}\|_0 \, \omega(d\mathbf{x}) \quad \text{s.t. } \mathbf{x} = \boldsymbol{\mu}_x[\mathbf{z}; \theta], \ \mathbf{z} = \boldsymbol{\mu}_z(\mathbf{x}; \phi) \ \forall \mathbf{x} \in \mathcal{X}. \tag{11}$$

In this way, the SAE model can be designed to enforce perfect reconstructions using the minimal number of nonzero latent dimensions. Such representations will generally correspond with the ground-truth manifold dimensions for data drawn according to Definition 4.1 along with Lipschitz continuous encoder and decoder networks $\boldsymbol{\mu}_z$ and $\boldsymbol{\mu}_x$. While conceptually notable, of course minimizing (11) is neither differentiable nor feasible to optimize, unlike our VAEase approach. We visualize SAE *approximate* solutions in Appendix A.3 below.

## A.3. Approximate Sparsity Using SAE-$\ell_1$ and SAE-log Models

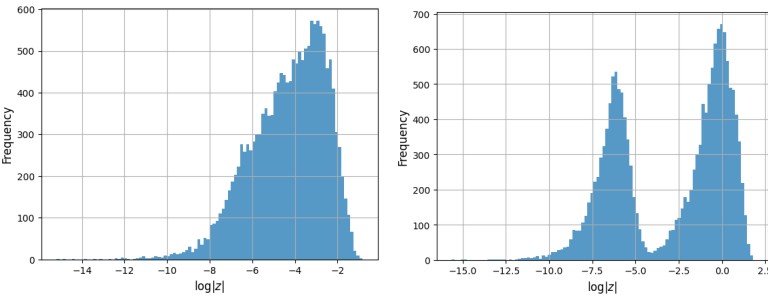

Figure 5: Because the ideal $\ell_0$ norm cannot be optimized via SGD, some form of smooth approximation is required by SAE models in practice. Here we show histograms of $|z|$ values (on a log scale) for SAE-$\ell_1$ (left) and SAE-log (right) on MNIST data. Because the weaker convex $\ell_1$ very loosely approximates the $\ell_0$ norm, there is no clear-cut partitioning between active (large values) and inactive (very small) dimensions. In contrast, the non-convex log-based penalty creates a clear separation. This separation is readily detectable by the variance criteria described in Section 5 as adopted in our experiments (applied in log-space).

# B. Further Analysis of VAE-based Models

This section provides more intuitive explanations for why VAEs struggle with adaptive sparsity, while VAEase does not.

## B.1. Why Fixed Sparsity in Existing VAE models

**Linear Decoder Case:** We illustrate the VAE tendency towards fixed sparsity using a linear decoder mean network given by $\boldsymbol{\mu}_x\big[\mathbf{z}; \theta\big] = \mathbf{W}\mathbf{z}$, with $\theta = \mathbf{W} \in \mathbb{R}^{d \times \kappa}$, analogous to the setup from Appendix A.1 although now reapplied to a new context. Granting this assumption and an arbitrary input sample $\mathbf{x}$, the VAE reconstruction loss will satisfy

$$-\mathbb{E}_{q_\phi(\mathbf{z}|\mathbf{x})}\big[\log p_\theta(\mathbf{x}|\mathbf{z})\big] \;\equiv\; \mathbb{E}_{q_\phi(\mathbf{z}|\mathbf{x})}\Big[\tfrac{1}{\gamma}\big\|\mathbf{x} - \mathbf{W}\mathbf{z}\big\|_2^2\Big] \;=\; \tfrac{1}{\gamma}\big\|\mathbf{x} - \mathbf{W}\boldsymbol{\mu}_z(\mathbf{x};\phi)\big\|_2^2 + \sum_{j=1}^{\kappa} \frac{\sigma_z(\mathbf{x};\phi)_j^2}{\gamma}\|\mathbf{w}_{:j}\|_2^2, \quad (12)$$

where $\mathbf{w}_{:j}$ denotes the $j$-th column of $\mathbf{W}$ and the first equivalence ignores constant factors independent of $\mathbf{W}$. As mentioned in Section 2.2 and proven in prior work, for inactive dimensions we have $q_\theta(z_j|\mathbf{x}) \approx \mathcal{N}(z_j|0,1)$ which implies $\sigma_z(\mathbf{x};\phi)_j \approx 1$. Hence the only way to avoid a divergent reconstruction loss as $\gamma \to 0$ is to have $\mathbf{w}_{:j} \to \mathbf{0}$ for *inactive* dimensions. In contrast, for an *active* dimension $j'$, recall that $q_\theta(z_{j'}|\mathbf{x}) \approx \mathcal{N}\big(z_{j'}|\mu_z(\mathbf{x};\phi)_{j'}, O(\gamma)\big)$. And so from (12) the factor

$$\frac{\sigma_z(\mathbf{x};\phi)_{j'}^2}{\gamma}\|\mathbf{w}_{:j'}\|_2^2 \;=\; \frac{O(\gamma)}{\gamma}\|\mathbf{w}_{:j'}\|_2^2 \;=\; O(1)\|\mathbf{w}_{:j'}\|_2^2 \tag{13}$$

will remain well-behaved even for $\mathbf{w}_{:j'} \neq \mathbf{0}$. And then of course since $\mathbf{W}$ is shared across all $\mathbf{x} \in \mathcal{X}$, zero- and nonzero-valued channels (i.e., columns of $\mathbf{W}$) will remain permanently so for all inputs leading to a fixed sparsity solution.

**Nonlinear Decoder Case:** When we generalize to nonlinear decoders, the problem raised above largely persists, although the situation becomes more nuanced. The added wrinkle in the nonlinear case is that a sufficiently complex decoder could in principle learn a form of partial adaptive sparsity pattern unlike in the linear setting, although with notable caveats that undermine practical realizations. Why is there such a gap between principle and practice here? To answer this question, it helps to contrast the core challenge facing an SAE decoder versus a VAE decoder.

Given a well-trained SAE model with suitable latent-space regularization, we may assume that $\mathbf{z} = \boldsymbol{\mu}_z(\mathbf{x};\phi)$ is a sparse vector in $\mathbb{R}^\kappa$. This implies that inactive dimensions can be trivially identified by simply checking which entries are equal to zero (or at least nearly so vis-à-vis a small threshold). Now consider a corresponding VAE model producing a latent representation with the exact same $\boldsymbol{\mu}_x(\mathbf{x};\phi)$, now serving as a posterior mean network, along with the posterior $\boldsymbol{\sigma}_z(\mathbf{x};\phi)$ variance satisfying

$$\sigma_z(\mathbf{x};\phi)_j \approx \left\{ \begin{array}{ll} 0, & j = \text{active dim} \\ 1, & j = \text{inactive dim} \end{array} \right. . \tag{14}$$

In this revised situation, what were once zero-valued inactive dimensions of $\mathbf{z}$ are now filled with samples from $\mathcal{N}(0, 1)$, and are thus much less distinguishable from the active dimensions. Figure 2 in the main text provides a toy illustration of this contrast. Because of this ambiguity, with limited capacity in the linear decoder setting from above, the only sure-fire way to differentiate active and inactive dimensions is to permanently encode a shared partition by zeroing out columns of $\mathbf{W}$. But in the nonlinear case there is another subtle option: One or more dimensions of $\mathbf{z}$ can be permanently designated as specialized active dimensions (just as in the linear case) but with an expanded, two-fold role that accommodates adaptive sparsity across the remaining decoder dimensions.

Specifically, to instantiate adaptive sparsity within a standard VAE architecture, there must exist a nonlinear decoder with $\rho \geq 1$ permanently active dimensions; the remaining $\kappa - \rho$ dimensions can then reflect adaptive sparsity assuming suitable decoder capacity. Of these $\rho$ permanently active dimensions, their role is two-fold:

1. Assist with reconstructions, i.e., the typical role of any ordinary active dimension, and

2. Designate which of the remaining $\kappa - \rho$ dimensions are active.

The latter is critical for instantiating adaptive sparsity, since otherwise it is not possible for the decoder to know with certainty which dimensions are active on an input-by-input basis. But of course actually implementing a decoder capable of inducing specialized dimensions that *simultaneously* fulfill the dual roles from above is quite challenging, and requires significant additional complexity, even if the data manifold(s) upon which $\mathbf{x} \in \mathcal{X}$ rest are relatively simple, e.g., a union of linear subspaces. And once such complexity is introduced, the risk of the VAE overfitting to finite-sample training data increases significantly. This is because, when granted sufficient decoder capacity, a VAE model applied to a finite sample training set can achieve perfect reconstructions using only a single active dimension, regardless of the true manifold structure (Dai et al., 2018)[Theorem 5]. Or stated differently, if VAE decoder capacity is expanded to instantiate adaptive sparsity, then this same capacity can be hijacked to trivially overfit to finite sample datasets even when the underlying manifold structure is relatively simple. We discuss how the VAEase architecture avoids this conundrum in Appendix B.3.

## B.2. Can Self-Attention Boost Linear Cases?

In Appendix B.1 we demonstrated why VAEs with the linear decoder $\boldsymbol{\mu}_x[\mathbf{z}; \theta] = \mathbf{W}\mathbf{z}$ will necessarily lead to fixed-sparsity representations. This is the same decoder assumed by recent models of LLM activation patterns adopted for interpretability purposes (Cunningham et al., 2024). Therefore as things presently stand, a VAE is *not* suitable for handling LLM activation data where adaptive sparse representations are required (indeed this explains the poor VAE performance on the LLM data from Table 3. However, it is still reasonable to ask, what if we were to modify the VAE encoder to

$$\boldsymbol{\mu}_x[\mathbf{z}; \theta] = \mathbf{W}\tau(\mathbf{z}), \tag{15}$$

where $\tau : \mathbb{R}^\kappa \mapsto \mathbb{R}^\kappa$ is an arbitrary parameterized function, e.g., a self-attention layer capable of learning adaptive sparse combinations of the columns of $\mathbf{W}$. In this way we could presumably retain the interpretability provided by the original linear decoder basis vectors, while avoiding rigid, fixed-sparsity solutions. Unfortunately though, this approach will be ineffectual because there can exist an infinite space of VAE global minimizers whereby $\tau(\mathbf{z})$ is *not* sparse. The reason is that VAE sparsity is induced by KL regularization as applied to the posterior mean and variance of $\mathbf{z}$, but $\tau$ operates *solely within the decoder where no regularization effect of any kind exists*. Hence even if the informative dimensions of $\mathbf{z}$ are sparse, $\tau(\mathbf{z})$ can map them to an arbitrary *non-sparse* representation that nonetheless minimizes the reconstruction error and globally minimizes the overall VAE loss. And provided that $\mathbf{W}$ is overcomplete, which it is by design for LLM interpretability applications, it will necessarily have a non-trivial null space such that infinite solutions with zero reconstruction error are possible.

## B.3. How VAEase Circumvents Fixed Sparsity

To provide more intuition into how VAEase avoids producing fixed sparsity patterns, we extend our examination of the linear decoder setting from Appendix B.1. Under these circumstances, the VAEase version of (12) is modified to

$$-\mathbb{E}_{q_\phi(\mathbf{z}|\mathbf{x})}\big[\log p_\theta(\mathbf{x}|\mathbf{z})\big] = \frac{1}{\gamma}\big\|\mathbf{x} - \mathbf{W}\boldsymbol{\mu}_z(\mathbf{x}; \phi)\big\|_2^2 + \sum_{j=1}^{\kappa} \frac{\sigma_z(\mathbf{x}; \phi)_j^2(1 - \sigma_z(\mathbf{x}; \phi)_j^2)}{\gamma}\|\mathbf{w}_{:j}\|_2^2. \tag{16}$$

From this expression we observe that the $\boldsymbol{\sigma}_z$-dependent regularization factor can now be minimized when *either* $\sigma_z(\mathbf{x}; \phi)_j^2 \to$ 1 for an inactive dimension, *or* $\sigma_z(\mathbf{x}; \phi)_j^2 \to 0$ for an active dimension. This allows all columns of $\mathbf{W}$ to potentially remain nonzero which then accommodates input-adaptive sparse solutions. This same strategy naturaly transitions seamlessly to nonlinear decoders as well, since inactive dimensions will automatically compress noise to zero regardless of the decoder structure.

Before concluding this section, it is worth considering a possible alternative way of achieving adaptive sparsity. Specifically, suppose instead of the VAEase technique of swapping $\widetilde{\mathbf{z}}$ for $\mathbf{z}$, we simply concatenate $\boldsymbol{\sigma}_z$ along with $\mathbf{z}$ to form the decoder function $\boldsymbol{\mu}_x\big([\mathbf{z} \; ; \; \boldsymbol{\sigma}_z(\mathbf{x}; \phi)]; \theta\big)$. As it turns out though, this strategy is not viable, since now the decoder can just learn to complete ignore $\mathbf{z}$ and use $\boldsymbol{\sigma}_z$ directly for input reconstruction. This will incur only a $O(1)$ penalty cost from the KL loss term, while allowing for negligible reconstruction error as measured by (5), provided $\kappa$ is sufficiently large. The associated $\log \gamma$ normalization factor (as shared by VAE models) will then tend towards minus infinity, pushing the overall loss towards minus infinity. Moreover, the KL loss applied in this way has no favoritism towards sparse solutions, and so the whole enterprise will not be successful in aligning with manifold structure.

## C. Exploring Data Label Alignment with VAEase Active Dimensions

In following a manifold hypothesis for high-dimensional data (Bengio et al., 2013), numerous studies have focused on estimating manifold dimensions (Ansuini et al., 2019; Brehmer & Cranmer, 2020; Mathieu & Nickel, 2020; Caterini et al., 2021). Along these lines it has been pointed out that the intrinsic dimension of a dataset (akin to the manifold dimension under the manifold assumption), rather than the ambient dimension, plays a more crucial role in model fitting (Fefferman et al., 2016; Pope et al., 2021). Subsequently, a union of manifolds hypothesis has been introduced to extend the original manifold hypothesis (Brown et al., 2022); this extension aligns more closely with intuitions regarding real-world labeled datasets. Moreover, companion studies indicate that local intrinsic dimension of data points may vary across datasets, naturally so for points belonging to different classes.

Conceptually, data points with the same label likely exhibit inherent consistency in associated features, although assuming that their sample set forms a single manifold is unlikely to hold in general. Thus, following (Brown et al., 2022), we use the VAEase models trained in Section 5.2 to estimate the active dimensions of the MNIST and FashionMNIST datasets, and group the data by labels. We then compute the average differences in active dimensions within and between groups, with the results presented in Table 5. The difference is computed as $|A \cup B| - |A \cap B|$ on two sets $A$ and $B$.

Table 5: Average differences in VAEase active dimensions within and between groups defined by class labels.

| datasets | Intra-class | Inter-class |
|---|---|---|
| MNIST | 0.3091 | 0.4861 |
| FashionMNIST | 0.6761 | 2.1143 |

The result in the Table 5 suggests that the labels in MNIST bear some significance w.r.t. partitioning of the manifolds; even more so for FashionMNIST. This is despite the fact that model training is completely independent of the class labels themselves.

## D. Experimental Settings

### D.1. 1D Local Minima Plots from Figure 3

For SAE, we instantiate the energy from ((1) using $\lambda_1 = \lambda_2 = 0.1$ and $h(z) = \log(|z| + \epsilon)$ where $\epsilon = 1e - 4$. For VAEase we fix $\gamma = 0.1$. We treat $\mu_z$ as a free parameter for both models; likewise for $\sigma_z$ as required by VAEase. For the decoders we adopt $\mu_x = wz$, where $w$ is a learnable parameter that has been optimized out of both SAE and VAEase models prior to plotting the figures.

### D.2. Synthetic Datasets

**Data generation.** For the MLP dataset, the data in each manifold are mapped by a MLP containing 2 linear layers with activation leaky-ReLU. Specifically, the input dimension of the MLPs are $r_i$ aligned to the respective manifold dimensions stated in main text. The dimensions of hidden layers follow the same $r_i$. The output dimensions are $d$ same as the ambient

dimension. The slope of leaky-ReLU is 0.2.

**Model architectures and training.** For the linear dataset, the encoders contain 4 linear layers connected by the activation function *Swish*, and the decoders are a single linear layer. The hidden dimension in encoders is set as $2\kappa$. For the MLP dataset, the encoders contain 3 residual blocks and each block consists 3 linear layers, and the decoders are 2 layer MLPs activated by leakyReLU with slope 0.2. The hidden dimension of residual blocks is $8\kappa$. Models were trained for 150 epochs on linear dataset and 310 epochs on the MLP dataset. The batch size was set to 1024. We choose $\kappa = 20$ for the linear dataset and $\kappa = 60$ for the MLP dataset. The learning rate for VAE models was 0.01 on linear dataset and 0.005 on the MLP dataset, while the learning rates for SAE models are 0.002 on linear dataset and 0.005 on the MLP dataset. The optimizer is Adam and learning rate scheduler is *CosineAnnealingWarmRestarts* with $T_0 = 10$. The penalty weights for SAE models in (1) are $\{\lambda_1 = 1e-3, \lambda_2 = 1e-4\}$ on linear dataset. On the MLP dataset, the weights are $\{\lambda_1 = 5e-4, \lambda_2 = 1e-5\}$ and $\{\lambda_1 = 5e-6, \lambda_2 = 1e-5\}$ for SAE-$\ell_1$ and SAE-log models. These hyperparameters are chosen to produce sufficiently small reconstruction errors ($1e-3$) while still enforcing sparsity as needed to facilitate head-to-head comparison with VAEs (which also produce negligible reconstruction error without any loss tuning parameters, i.e., $\gamma$ is learned during training).

### D.3. Real-World Datasets

**MNIST and FashionMNIST.** Model encoders for the MNIST dataset contain 1 input convolution layer and 2 residual blocks, where each block has 2 convolution layers. $\kappa$ is 32. The decoders consist of 1 *DenseNN*, 1 upsampling layer, 2 residual blocks and a convolution layer. The residual block here also has 2 convolution layers. Models were trained for 300 epochs in the MNIST dataset using batch size 2048. Learning rates are 0.05. The optimizer is Adam and learning rate scheduler is *CosineAnnealingWarmRestarts* with $T_0 = 20$. The penalty weights are $\{\lambda_1 = 5e-1, \lambda_2 = 1e-1\}$ and $\{\lambda_1 = 1e-2, \lambda_2 = 5e-2\}$ for SAE-$\ell_1$ and SAE-log models on MNIST dataset, $\{\lambda_1 = 2e-2, \lambda_2 = 2e-2\}$ and $\{\lambda_1 = 1.3e-2, \lambda_2 = 2e-2\}$ on FashionMNIST dataset correspondingly. Additionally, the estimated $|z|$ values follow a heavy-tailed distribution as shown in Figure 5, so it is more reasonable to apply the variance criteria w.r.t. $\log|z|$ for dividing active and inactive dimensions.

**LLM intermediate-layer activations.** As for data, online code is available for gathering the intermediate activation layers.[7] For model architectures, the encoders for LLM dataset consist of only one linear layer and a ReLU layer, and the decoders are simply linear transforms consistent with popular use cases (Chaudhary & Geiger, 2024). For VAE models, there is also a linear layer as needed to output $\boldsymbol{\sigma}_z$. Models were trained for 35 epochs with batch-size 2048. The learning rate is 0.005 and $\kappa$ is 300. The learning rate scheduler is *CosineAnnealingWarmRestarts* with $T_0 = 5$. The penalty weights on this dataset are $\{\lambda_1 = 3.2e-1, \lambda_2 = 2e-1\}$ and $\{\lambda_1 = 1e-2, \lambda_2 = 2e-1\}$ for SAE-$\ell_1$ and SAE-log models, respectively, tuned for similar reconstruction error.

**Yelp text embedding dataset.** The Yelp dataset was obtained from hugging face.[8] The model architectures used for these experiments are the same as those from above as applied LLM intermediate activations. Models were trained for 150 epochs with a batch size of 512. The learning rate is 0.001 and $\kappa$ is 512. The learning rate scheduler is *CosineAnnealingWarmRestarts* with $T_0 = 10$. The penalty weights are $\{\lambda_1 = 3e-3, \lambda_2 = 1e-2\}$ and $\{\lambda_1 = 2e-4, \lambda_2 = 5e-4\}$ for SAE-$\ell_1$ and SAE-log models, respectively, tuned for similar reconstruction error as in previous experiments.

**Pesudo-MNIST.** We trained a GAN on MNIST for 500 epochs using publicly-available code.[9] Subsequently we generated 500,000 new samples to form the pseudo-MNIST dataset. We adopted the same encode/decoder architecture for VAEase as used for the original MNIST data. We then trained VAEase model for 35 epochs on this dataset using batch size 512. $\kappa$ is 100 and the learning rate is 0.005. The optimizer is Adam and learning rate scheduler is *CosineAnnealingWarmRestarts* with $T_0 = 50$. For the diffusion model, we adopted the code[10] associated with Stanczuk et al. (2024) and trained for 70 epochs. The NB post-processing manifold dimension estimator is also implemented using the same repo. Meanwhile, the FLIPD estimator originates from a separate repo.[11]

---

[7]https://github.com/HoagyC/sparse_coding

[8]https://huggingface.co/datasets/rmovva/HypotheSAEs

[9]https://github.com/eriklindernoren/PyTorch-GAN

[10]https://github.com/GBATZOLIS/ID-diff

[11]https://github.com/layer6ai-labs/diffusion_memorization/blob/main/flipd_utils.py

### D.4. Representative Samples from Pseudo-MNIST

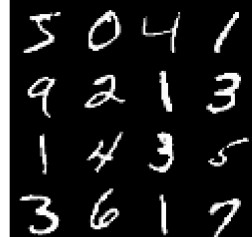 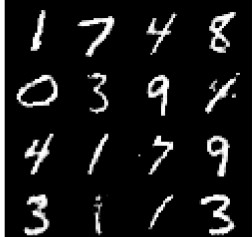

Figure 6: *Samples from Pesudo-MNIST*. On the left we show samples from the original MNIST, while on the right are samples from the proposed pesudo-MNIST; the latter are generated by a GAN trained on MNIST with a known upper bound on the manifold dimension. Because visually the two sets of samples are quite similar, it increases our credence that the pseudo-MNIST data manifold dimension is a reasonable approximation to that of the original MNIST.

## E. Technical Proofs

### E.1. Proof of Theorem 4.5

We first give a feasible solution and compute the loss rate, which could be the upper rate bound for the optimal loss. Then we prove that there is no solution to achieve a lower loss bound, and derive necessary conditions for the rate-optimal solution by analyzing the order of each term in the loss function.

**A feasible solution and upper bound.** First, we will construct a solution to give a upper bound of optimal loss rate. For any manifold $\mathcal{M}_i \subseteq \mathcal{M}$, there exists an $L$-Lipschitz invertible function $\psi_i(\cdot)$ that maps $\mathcal{M}_i$ to $\mathbb{R}^{r_i}$ and $\psi_i^{-1}(\mathbf{0}) = \mathbf{0}$. Then we can construct $\boldsymbol{\mu}_z, \boldsymbol{\Sigma}_z \stackrel{\text{def}}{=} \text{diag}[\boldsymbol{\sigma}_z^2]$ and $\boldsymbol{\mu}_x$ s follows:

$$\boldsymbol{\mu}_z = (I(\mathbf{x} \in \mathcal{M}_1)\psi_1(\mathbf{x})^\top/(1 - \gamma^{1/2}), \ldots, I(\mathbf{x} \in \mathcal{M}_n)\psi_n(\mathbf{x})^\top/(1 - \gamma^{1/2}), \mathbf{0}^\top)^\top,$$

$$\boldsymbol{\Sigma}_z = \text{diag}\{(\mathbf{e}_{r_1}^\top - I(\mathbf{x} \in \mathcal{M}_1)(1 - \gamma)\mathbf{e}_{r_1}^\top, \ldots, \mathbf{e}_{r_n}^\top - I(\mathbf{x} \in \mathcal{M}_n)(1 - \gamma)\mathbf{e}_{r_n}^\top, \mathbf{e}_{\kappa - \sum r_i}^\top)^\top\},$$

$$\boldsymbol{\mu}_x = \sum_i \psi_i^{-1}(\{(\mathbf{I} - \boldsymbol{\Sigma}_z^{1/2})\mathbf{z}\}_{\sum_{j=1}^{i-1} r_j : \sum_{j=1}^{i-1} r_j + r_i}),$$

where $\mathbf{0}$ is an all-zero vector of dimension $\kappa - \sum_i r_i$ and $\mathbf{e}_r$ denotes an all-ones vector of dimension $r$. As the sum of multiple $L$-Lipschitz functions, $\boldsymbol{\mu}_x$ is naturally also $L$-Lipschitz on set $\mathcal{M}$, satisfying Definition 4.2. Now compute the corresponding loss function to check if it reaches the lower bound we discussed.

$$\int_{\mathcal{M}} \mathbf{E}_{q_\phi(\mathbf{z}|\mathbf{x})}[\|\mathbf{x} - \mu_x\|^2]\omega(d\mathbf{x}) = \sum_i \int_{\mathcal{M}_i} \mathbf{E}_{q_\phi(\mathbf{z}|\mathbf{x})}[\|\psi_i^{-1}(\psi_i(\mathbf{x})) - \psi_i^{-1}(\psi_i(\mathbf{x}) + (1 - \gamma^{1/2})\gamma^{1/2}\boldsymbol{\varepsilon}_i)\|^2]\omega(d\mathbf{x})$$

$$\leq \sum_i \int_{\mathcal{M}_i} L^2 \mathbf{E}_{\boldsymbol{\varepsilon}_i}[\|\psi_i(\mathbf{x}) - \psi_i(\mathbf{x}) + (1 - \gamma^{1/2})\gamma^{1/2}\boldsymbol{\varepsilon}_i\|^2]\omega(d\mathbf{x})$$

$$= \sum_i \int_{\mathcal{M}_i} L^2(1 - \gamma^{1/2})^2 \gamma r_i \omega(d\mathbf{x}) = L^2 \sum_i (1 - \gamma^{1/2})^2 \gamma r_i \alpha_i$$

Then the KL term can be computed as

$$\int_{\mathcal{M}} (\text{tr}(\boldsymbol{\Sigma}_z) - \log|\boldsymbol{\Sigma}_z| + \boldsymbol{\mu}_z^\top \boldsymbol{\mu}_z - \kappa)\omega(d\mathbf{x})$$

$$= \sum_i \int_{\mathcal{M}_i} (\gamma r_i + (\kappa - r_i) - r_i \log\gamma + \boldsymbol{\mu}_z^\top \boldsymbol{\mu}_z - \kappa)\omega(d\mathbf{x})$$

$$\leq \sum_i \int_{\mathcal{M}_i} ((\gamma - 1)r_i - r_i \log\gamma + L^2 \mathbf{x}^\top \mathbf{x})\omega(d\mathbf{x})$$

$$= \sum_i -\alpha_i r_i \log\gamma + L^2 \int_{\mathcal{M}} \mathbf{x}^\top \mathbf{x}\omega(d\mathbf{x}) + O(1)$$

Hence the upper bound of optimal loss rate is $(d - \sum_i \alpha_i r_i) \log\gamma/2 + O(1)$, obtained by perfect reconstruction function $\mu_x, \mu_z$ and $r_i$ instances of $\sigma_{zj} = O(\gamma)$ for any $\mathbf{x}$.

Note that we construct this solution without explicitly considering the intersection of the manifolds for the following reasons. First, if the region of intersection is in a lower dimensional manifold, then according to Definition 4.1 it has zero probability measure and therefore contributes nothing to the overall loss. And secondly, if the intersection dimension matches the dimensions of the corresponding manifolds that are interesting, then these manifolds can be merged into a single manifold. Either way, we are able to integrate the loss on well-defined manifolds one by one.

**Optimal rate and associated necessary conditions.** We will now show that the above loss rate is also optimal and the active dimension is equal to the local manifold dimension almost sure. Consider the loss function in VAEase model,

$$\mathcal{L}_{\text{VAEase}}(\mathbf{x}; \phi, \theta, \gamma) = - \mathbf{E}_{q_\phi(\mathbf{z}|\mathbf{x})} [\log p_\theta(\mathbf{x}|\mathbf{z})] + \mathbf{KL}(q_\phi(\mathbf{z}|\mathbf{x})||p(\mathbf{z})),$$

$$= \frac{d}{2} \log(2\pi\gamma) + \mathbf{E}_{q_\phi(\mathbf{z}|\mathbf{x})} \left[\frac{\|\mathbf{x} - \boldsymbol{\mu}_x\|^2}{2\gamma}\right] + \frac{1}{2} \left(\text{tr}(\boldsymbol{\Sigma}_z) - \log|\boldsymbol{\Sigma}_z| + \boldsymbol{\mu}_z^\top \boldsymbol{\mu}_z - \kappa\right)$$

Assume there exists a solution that can achieve a rate smaller than $(d - \sum_i \alpha_i r_i) \log\gamma/2 + O(1)$. Since terms $\text{tr}(\boldsymbol{\Sigma}_z) + \boldsymbol{\mu}_z^\top \boldsymbol{\mu}_z - \kappa$ and reconstruction error are $O(1)$ in the shown solution, to reduce the order of the loss, one can only reduce the term $\log|\boldsymbol{\Sigma}_z|$.

**Optimal rate for active dimensions.** Suppose there are a set $\mathcal{X}' \subseteq \mathcal{M}_i$ satisfying that, for any $\mathbf{x} \in \mathcal{X}'$, $\log|\boldsymbol{\Sigma}_z| = r_i \Omega(-\log\gamma)$ and $\int_{\mathcal{X}'} \omega(d\mathbf{x}) > 0$. Then the number of active dimensions for $\mathbf{x}$ must be less than $r_i$. Suppose that there are $r_i - 1$ active dimensions for which $\sigma_z^2(\mathbf{x}; \phi_\gamma)_j = O(\gamma)$, and there is one dimension for which $\sigma_z^2(\mathbf{x}; \phi_\gamma)_j = O(g(\gamma)) > O(\gamma) \to 0$ as $\gamma \to 0$ (if there is no other dimension $j$ satisfying $\lim_{\gamma \to 0} \sigma_z^2(\mathbf{x}; \phi_\gamma)_j = 0$, we assert the reconstruction error would be $\Theta(1)$, resulting in infinite loss). We will show that the reconstruction error cannot remain at $O(\gamma)$ in this case.

Since $\mathcal{X}'$ is a subset of the manifold $\mathcal{M}_i$ and its measure is not zero, at least $r_i$ dimensions of information are needed to reconstruct $\mathcal{X}'$ with Lipschitz functions. Without loss of generality, we set the channels used for reconstruction to the first $r_i$ dimensions, i.e., $\boldsymbol{\mu}(\mathbf{x}) = \tilde{\boldsymbol{\mu}}(\mathbf{x}_{1:r_i})$. To get the lower reconstruction loss, the function $\tilde{\boldsymbol{\mu}}$ must map $\mathbb{R}^{r_i}$ to $\mathcal{X}'$. In addition, there exist a compact set $\mathcal{Z} \subseteq \mathbb{R}^{r_i}$ and a positive constant $l$ satisfying $\|\tilde{\boldsymbol{\mu}}(\mathbf{z}_1) - \tilde{\boldsymbol{\mu}}(\mathbf{z}_2)\| \geq l\|\mathbf{z}_1 - \mathbf{z}_2\|$ for any $\mathbf{z}_1, \mathbf{z}_2 \in \mathcal{Z}$.

Then we have

$$\int_{\mathcal{X}'} \mathbf{E}_{\boldsymbol{\varepsilon}\sim\mathcal{N}(0,\mathbf{I})}[\|\mathbf{x} - \boldsymbol{\mu}_x((\mathbf{I} - \boldsymbol{\Sigma}_z^{1/2})\boldsymbol{\mu}_z + (\mathbf{I} - \boldsymbol{\Sigma}_z^{1/2})\boldsymbol{\Sigma}_z^{1/2}\boldsymbol{\varepsilon})\|^2]\omega(d\mathbf{x})$$

$$\geq \int_{\mathcal{X}'} \mathbf{E}_{\boldsymbol{\varepsilon}\sim\mathcal{N}(0,\mathbf{I})}[\|\mathbf{x} - \boldsymbol{\mu}_x((\mathbf{I} - \boldsymbol{\Sigma}_z^{1/2})\boldsymbol{\mu}_z + (\mathbf{I} - \boldsymbol{\Sigma}_z^{1/2})\boldsymbol{\Sigma}_z^{1/2}\boldsymbol{\varepsilon})\|^2|\|\boldsymbol{\varepsilon}\|_\infty \leq 1]\omega(d\mathbf{x})$$

$$\geq \int_{\mathcal{X}'} \mathbf{E}_{\boldsymbol{\varepsilon}\sim\mathcal{N}(0,\mathbf{I})}[\|\mathbf{x} - \boldsymbol{\mu}_x((\mathbf{I} - \boldsymbol{\Sigma}_z^{1/2})\boldsymbol{\mu}_z + (\mathbf{I} - \boldsymbol{\Sigma}_z^{1/2})\boldsymbol{\Sigma}_z^{1/2}\boldsymbol{\varepsilon})\|^2|$$

$$\{\|\boldsymbol{\varepsilon}\|_\infty \leq 1\} \cap \{\mathcal{B}_1\left((\mathbf{I} - \boldsymbol{\Sigma}_z^{1/2})\boldsymbol{\mu}_z, (\mathbf{I} - \boldsymbol{\Sigma}_z^{1/2})\boldsymbol{\Sigma}_z^{1/2}\right) \subseteq \mathcal{Z}\}]\omega(d\mathbf{x})$$

$$\geq \int_{\mathcal{X}'} \int_{\mathbb{R}^\kappa} (l^2 \mathrm{tr}((\mathbf{I} - \boldsymbol{\Sigma}_z^{1/2})^2 \boldsymbol{\Sigma}_z \boldsymbol{\varepsilon}^2)) I(\{\|\boldsymbol{\varepsilon}\|_\infty \leq 1\} \cap \{\mathcal{B}_1\left((\mathbf{I} - \boldsymbol{\Sigma}_z^{1/2})\boldsymbol{\mu}_z, (\mathbf{I} - \boldsymbol{\Sigma}_z^{1/2})\boldsymbol{\Sigma}_z^{1/2}\right) \subseteq \mathcal{Z}\})\omega_N(d\boldsymbol{\varepsilon})\omega(d\mathbf{x})$$

$$\geq \int_{\mathcal{X}'} c_1(\sum_i \boldsymbol{\sigma}_z^2(\mathbf{x};\phi_\gamma)_j(1 - \boldsymbol{\sigma}_z(\mathbf{x};\phi_\gamma)_j)^2) I(\mathcal{B}_1\left((\mathbf{I} - \boldsymbol{\Sigma}_z^{1/2})\boldsymbol{\mu}_z(\mathbf{x}), (\mathbf{I} - \boldsymbol{\Sigma}_z^{1/2})\boldsymbol{\Sigma}_z^{1/2}\right) \subseteq \mathcal{Z})\omega(d\mathbf{x}) \tag{17}$$

$$\geq c_1 O(g(\gamma)) \int_{\mathcal{X}'} \omega(d\mathbf{x}),$$

where $\mathcal{B}_1((\mathbf{I} - \boldsymbol{\Sigma}_z^{1/2})\boldsymbol{\mu}_z(\mathbf{x}), (\mathbf{I} - \boldsymbol{\Sigma}_z^{1/2})\boldsymbol{\Sigma}_z^{1/2})$ is a $l_1$ norm ball and $\int_{\mathcal{M}_i} I(\mathcal{B}_1((\mathbf{I} - \boldsymbol{\Sigma}_z^{1/2})\boldsymbol{\mu}_z(\mathbf{x}), (\mathbf{I} - \boldsymbol{\Sigma}_z^{1/2})\boldsymbol{\Sigma}_z^{1/2}) \subseteq \mathcal{Z})\omega(d\mathbf{x})$ could go to $\alpha_i$ when $l$ is sufficiently small. The total loss on set $\mathcal{X}'$ should be in order $\{O(g(\gamma))/\gamma + d\log\gamma - (r_i - 1)\log\gamma - \log g(\gamma) + O(1)\} \int_{\mathcal{X}'} \omega(d\mathbf{x})/2$. Since $O(g(\gamma))/\gamma - \log g(\gamma) \geq O(1) + \log\gamma$, there is no lower order for the feasible loss for any $g(\gamma) > O(\gamma)$. However, the equation also shows that when $\int_{\mathcal{X}'} \omega(d\mathbf{x}) = \gamma/O(g(\gamma)) \to 0$, the loss difference could be absorbed by $O(1)$. In other words, the number of active dimensions in $\mathbf{x}$ will match the manifold dimensions, with a probability of almost 1 as $\gamma \to 0$.

**Optimal rate for inactive dimensions.** First, from the above analysis, the rates of variance on active dimensions converge to 0. In contrast, the rate for inactive dimensions should be $\Theta(1)$. However, it is not sufficient to ensure the effectiveness of our revamping mechanism, which is expected as $1 - \sigma_j \approx 0$ for inactive dimensions. Formula 17 implies the rate requirement of $O(\gamma)$ reconstruction loss on inactive dimensions.

Suppose that the inactive dimension $j^*$ has no contribution in function $\boldsymbol{\mu}(\cdot)$ on set $\mathcal{Z}^*$, which means $\boldsymbol{\mu}(\mathbf{x}) = \boldsymbol{\mu}(\mathbf{x}')$ for all $(\mathbf{x}, \mathbf{x}')$ and $\mathbf{x}, \mathbf{x}' \in \mathcal{Z}^*$ are only different at dimension $j^*$. Then the reconstruction loss is independent with $\sigma_j^2$ and the left relative terms in loss function are $\sigma_j^2 - \log\sigma_j^2$ optimized at $\sigma_j = 1$.

Otherwise, with some constant $C$, consider that there exists a subset $\tilde{\mathcal{Z}}_{i'} \subseteq \mathcal{Z}_{i'} = \{\mathbf{z}|\mathbf{z} = (\mathbf{I} - \boldsymbol{\Sigma}_z^{1/2})\mathbf{z}' + (\mathbf{I} - \boldsymbol{\Sigma}_z^{1/2})\boldsymbol{\Sigma}_z^{1/2}\boldsymbol{\varepsilon}, \mathbf{z}' = \boldsymbol{\mu}_z(\mathbf{x}'), \mathbf{x}' \in \mathcal{M}_{i'}, \|\boldsymbol{\varepsilon}\|_\infty \leq C\}$, where an inactive dimension $j^*$ contributes to function $\boldsymbol{\mu}_x(\cdot)$, also $\partial\boldsymbol{\mu}_x(\mathbf{z})/\partial z_{j^*} \neq 0$. Combined with the continuity assumption, we could obtain a subset $\tilde{\mathcal{Z}}_{i'}(l') \subseteq \tilde{\mathcal{Z}}_{i'}$ that $\|\boldsymbol{\mu}_x(\mathbf{z}) - \boldsymbol{\mu}_x(\mathbf{z}')\| \geq l'|z_{j^*} - z'_{j^*}|$ for $\mathbf{z}, \mathbf{z}' \in \tilde{\mathcal{Z}}$. Let $1 - \sigma_{j^*} = g'(\gamma) \to 0$, we then have a similar analysis in formula 17 as

$$\int_{\mathcal{M}_{i'}} \mathbf{E}_{\boldsymbol{\varepsilon}\sim\mathcal{N}(0,\mathbf{I})}[\|\mathbf{x} - \boldsymbol{\mu}_x((\mathbf{I} - \boldsymbol{\Sigma}_z^{1/2})\boldsymbol{\mu}_z + (\mathbf{I} - \boldsymbol{\Sigma}_z^{1/2})\boldsymbol{\Sigma}_z^{1/2}\boldsymbol{\varepsilon})\|^2]\omega(d\mathbf{x})$$

$$\geq \int_{\mathcal{M}_{i'}} \mathbf{E}_{\boldsymbol{\varepsilon}\sim\mathcal{N}(0,\mathbf{I})}[\|\mathbf{x} - \boldsymbol{\mu}_x((\mathbf{I} - \boldsymbol{\Sigma}_z^{1/2})\boldsymbol{\mu}_z + (\mathbf{I} - \boldsymbol{\Sigma}_z^{1/2})\boldsymbol{\Sigma}_z^{1/2}\boldsymbol{\varepsilon})\|^2|\|\boldsymbol{\varepsilon}\|_\infty \leq C]\omega(d\mathbf{x})$$

$$\geq \int_{\mathcal{M}_{i'}} \mathbf{E}_{\boldsymbol{\varepsilon}\sim\mathcal{N}(0,\mathbf{I})}[\|\mathbf{x} - \boldsymbol{\mu}_x((\mathbf{I} - \boldsymbol{\Sigma}_z^{1/2})\boldsymbol{\mu}_z + (\mathbf{I} - \boldsymbol{\Sigma}_z^{1/2})\boldsymbol{\Sigma}_z^{1/2}\boldsymbol{\varepsilon})\|^2|$$

$$\{\|\boldsymbol{\varepsilon}\|_\infty \leq C\} \cap \{(\mathbf{I} - \boldsymbol{\Sigma}_z^{1/2})\boldsymbol{\mu}_z + (\mathbf{I} - \boldsymbol{\Sigma}_z^{1/2})\boldsymbol{\Sigma}_z^{1/2}\boldsymbol{\varepsilon} \in \tilde{\mathcal{Z}}_{i'}(l')\}]\omega(d\mathbf{x})$$

$$\geq \int_{\mathcal{M}_{i'}} \int_{\|\boldsymbol{\varepsilon}\|_\infty \leq C} l'^2 (1 - \sigma_{j^*})^2 \sigma_{j^*}^2 \varepsilon_{j^*}^2 I(\{(\mathbf{I} - \boldsymbol{\Sigma}_z^{1/2})\boldsymbol{\mu}_z + (\mathbf{I} - \boldsymbol{\Sigma}_z^{1/2})\boldsymbol{\Sigma}_z^{1/2}\boldsymbol{\varepsilon} \in \tilde{\mathcal{Z}}_{i'}(l')\})\omega_N(d\boldsymbol{\varepsilon})\omega(d\mathbf{x})$$

$$\geq c_2 l'^2 g'(\gamma)^2 \int_{\mathcal{M}_{i'}} \int_{\|\boldsymbol{\varepsilon}\|_\infty \leq C} I(\{(\mathbf{I} - \boldsymbol{\Sigma}_z^{1/2})\boldsymbol{\mu}_z + (\mathbf{I} - \boldsymbol{\Sigma}_z^{1/2})\boldsymbol{\Sigma}_z^{1/2}\boldsymbol{\varepsilon} \in \tilde{\mathcal{Z}}_{i'}(l')\})\omega_N(d\boldsymbol{\varepsilon})\omega(d\mathbf{x}).$$

Denote the last integral terms by $\mathcal{A}_{i'}(C, l')$, then is is held that $\mathcal{A}_{i'}(C, l') \to \int_{\mathcal{M}_{i'}} I(\mathbf{z} \in \tilde{\mathcal{Z}}_{i'}) \omega(d\mathbf{x})$ when $l'$ is sufficiently small. To make sure the reconstruction loos is $O(\gamma)$, we could obtain that if $\mathcal{A}_{i'}(C, l') \geq c > 0$ as $\gamma \to 0$, $g'(\gamma) = O(\gamma^{1/2})$ must hold; if $\mathcal{A}_{i'}(C, l') \to 0$, $g'(\gamma) = O(\gamma^{1/2})$ could be violated almost with a probability of 0.

Since $\int_{\mathcal{M}_{i'}} I(\mathbf{z} \in \mathcal{Z}_{i'}) \omega(d\mathbf{x}) \to \int_{\mathcal{M}_{i'}} \omega(d\mathbf{x})$ with sufficiently large $C$ and $\mathcal{Z}_{i'} - \tilde{\mathcal{Z}}_{i'} \to \mathcal{Z}^*$ is the set where dimension $j^*$ contributes nothing to $\boldsymbol{\mu}_x(\cdot)$, we finalize the proof that $\sigma_{j^*}^2 = 1 - O(\gamma)$ holds almost with a probability of 1.

$\square$

### E.2. Proof of Corollary 4.6

The proof is related to that of Theorem 4.5, but the upper bound of the loss differs. Without the additional $(1 - \sigma_z)$ to set inactive dimensions to zero, the feasible solution in VAEase model is invalid. Here we consider a simple dataset combined with a point dataset and a line dataset, and $\kappa = 1$ is the dimension of latent variables. Assume there exists a solution satisfying

$$\int_{\mathcal{M}_i} I(|\mathcal{A}(x)| = r_i) \omega(d\mathbf{x}) = \alpha_i + o(1).$$

To avoid the loss function diverge to $+\infty$, a necessary condition is $\mathcal{R} = O(\gamma)$. For the point manifold, there should be no active dimensions. Since $\mathcal{M}_0$ only contains one point $\mathbf{x}_0$, to make the reconstruction error is $O(\gamma)$, it leads to $P(\|\boldsymbol{\mu}_x(z) - \mathbf{x}_0\| \geq 1) \leq \mathcal{R} = O(\gamma)$ by Chebyshev's inequality. For the line manifold, it should be held that $\mathbf{x} = \boldsymbol{\mu}_x(\mu_z(\mathbf{x})) + O(\sqrt{\gamma})$.

Assume the point manifold is on the line and consider the data point around $\mathbf{x}_0$. For the a compact set $\mathcal{X} := \{\mathbf{x}|\|\mathbf{x} - \mathbf{x}_0\| \geq 1\}$, we know preimage of set $\mathcal{X}$ under Lipschitz mapping $\mu_x$ have probability no more than $O(\gamma)$ under the distribution $\mathcal{N}(\mu_z(\mathbf{x}_0), 1)$. This means that the preimage $\mathcal{Z}$, as a compact set on $\mathbb{R}$, must satisfy

$$\int_{\mathcal{Z}} \exp(-(z - \mu_z(\mathbf{x}_0))^2/2) dz = O(\gamma). \tag{18}$$

Due to the Lipschitz continuity of $\mu_z$, consider $\mathbf{x}_0$ and $\mathbf{x}'$ satisfying $\|\mathbf{x}' - \mathbf{x}_0\| = 1$, then $|\mu_z(\mathbf{x}') - \mu_z(\mathbf{x}_0)|$ should be bounded by some constant. Then compact set $\mathcal{Z}$ should be small than $O(\gamma)$ under the Lagrangian measure to keep (E.2) fixed. Then there exist a $\mathbf{x}'' \in \mathcal{X}$ satisfying $\|\mathbf{x}'' - \mathbf{x}'\| > c$ for some positive constant $c$. So we finally get $\|\boldsymbol{\mu}_x(\mu_z(\mathbf{x}'')) - \boldsymbol{\mu}_x(\mu_z(\mathbf{x}'))\| \leq L\|\mu_z(\mathbf{x}'') - \mu_z(\mathbf{x}')\|$ because the decoder mean function is Lipschitz. As a result, we obtain $L \geq (c - O(\sqrt{\gamma}))/O(\gamma)$ with probability almost 1, which means the decoder mean function is not Lipschitz almost surely. From this conflict, we can conclude that there is no solution that satisfies adaptive sparsity for VAE on this dataset. $\square$

### E.3. Proof of Theorem 4.7

Since the $\mathbf{U}$ is known, the optimization problem can be conveniently decoupled into $d$ independent one-dimensional optimization problems (this stems from the fact that the Frobenius norm is invariant to orthonormal transformations). So we just need to verify that there is a unique minima for each one-dimensional problem. The loss function for single point in VAEase model is

$$\mathcal{L}_{\text{VAEase}}(x; w, \mu_z, \sigma_z^2) = \{d \log(2\pi\gamma) + \|x - w(1 - \sigma_z)\mu_z\|_2^2/\gamma + w^2\sigma_z^2(1 - \sigma_z)^2/\gamma + \sigma_z^2 - \log \sigma_z^2 + \mu_z^2 - 1\}/2$$

and the partial derivatives are

$$\frac{\partial 2\mathcal{L}_{\text{VAEase}}(x; w, \mu_z, \sigma_z^2)}{\partial w} = 2(1 - \sigma_z)\mu_z(w(1 - \sigma_z)\mu_z - x)/\gamma + 2w\sigma_z^2(1 - \sigma_z)^2/\gamma,$$

$$\frac{\partial 2\mathcal{L}_{\text{VAEase}}(x; w, \mu_z, \sigma_z^2)}{\partial \sigma_z} = -2w\mu_z(w(1 - \sigma_z)\mu_z - x)/\gamma + 2w^2\sigma_z(1 - \sigma_z)^2/\gamma + 2w^2\sigma_z^2(\sigma_z - 1)/\gamma + 2\sigma_z - 2/\sigma_z,$$

$$\frac{\partial 2\mathcal{L}_{\text{VAEase}}(x; w, \mu_z, \sigma_z^2)}{\partial \mu_z} = 2w(1 - \sigma_z)(w(1 - \sigma_z)\mu_z - x)/\gamma + 2\mu_z.$$

From the KKT condition, we can obtain

$$w = \frac{\mu_z x}{(1 - \sigma_z)(\sigma_z^2 + \mu_z^2)},$$

$$\frac{w^2(1 - \sigma_z)}{\gamma}(\sigma_z(1 - 2\sigma_z) - \mu_z^2) + \frac{w\mu_z x}{\gamma} = \sigma_z^{-1} - \sigma_z,$$

$$\mu_z = \frac{w(1 - \sigma_z)x}{w^2(1 - \sigma_z)^2 + \gamma}.$$

Substituting the first equation into the last equation, we obtain

$$\mu_z = \frac{\frac{\mu_z x}{(1 - \sigma_z)(\sigma_z^2 + \mu_z^2)}(1 - \sigma_z)x}{\frac{\mu_z^2 x^2}{(1 - \sigma_z)^2(\sigma_z^2 + \mu_z^2)^2}(1 - \sigma_z)^2 + \gamma}$$

$$\frac{\mu_z^2 x^2}{(\sigma_z^2 + \mu_z^2)^2} + \gamma = \frac{x^2}{(\sigma_z^2 + \mu_z^2)},$$

$$\mu_z^2 x^2 + \gamma(\sigma_z^2 + \mu_z^2)^2 = x^2(\sigma_z^2 + \mu_z^2),$$

$$\gamma(\sigma_z^2 + \mu_z^2)^2 = x^2 \sigma_z^2.$$

Finally, we compute the second equation as

$$\frac{\mu_z^2 x^2}{(1 - \sigma_z)^2(\sigma_z^2 + \mu_z^2)^2}(1 - \sigma_z)(\sigma_z(1 - 2\sigma_z) - \mu_z^2) + \frac{\mu_z x}{(1 - \sigma_z)(\sigma_z^2 + \mu_z^2)}\mu_z x = \gamma(\sigma_z^{-1} - \sigma_z),$$

$$\frac{\mu_z^2 x^2}{(\sigma_z^2 + \mu_z^2)^2}(\sigma_z(1 - 2\sigma_z) - \mu_z^2) + \frac{\mu_z^2 x^2}{(\sigma_z^2 + \mu_z^2)} = \gamma(\sigma_z^{-1} - \sigma_z)(1 - \sigma_z),$$

$$\frac{\mu_z^2 x^2}{(\sigma_z^2 + \mu_z^2)^2}(\sigma_z(1 - 2\sigma_z) - \mu_z^2 + \sigma_z^2 + \mu_z^2) = \gamma(\sigma_z^{-1} - \sigma_z)(1 - \sigma_z),$$

$$\frac{\mu_z^2 x^2}{(\sigma_z^2 + \mu_z^2)^2} = \gamma(\sigma_z^{-2} - 1).$$

Using $(\sigma_z^2 + \mu_z^2)^2 = x^2\sigma_z^2/\gamma$, we have

$$\frac{\mu_z^2 x^2}{x^2\sigma_z^2/\gamma} = \gamma(\sigma_z^{-2} - 1),$$

$$\mu_z^2 = (1 - \sigma_z^2).$$

Substituting this expression back into the above result, we get $\gamma = x^2\sigma_z^2$ and $\sigma_z^2 = \gamma/x^2$. At this point, we have calculated the unique minimum point $(\mu_z^*, \sigma_z^*) = (\sqrt{1 - \gamma/x^2}, \sqrt{\gamma/x^2})$ for the one-dimensional problem. The multi-dimensional conclusion then naturally follows. $\qquad\square$

### E.4. Proof of Corollary 4.8

Following the same approach as in the previous proof above, we actually only need to demonstrate that a one-dimensional problem has two local minima. The loss function for single point in SAE model is

$$\mathcal{L}_{\text{SAE}}(x; w, z) = (x - wz)^2 + \lambda_1 h(z) + \lambda_2 w^2,$$

and the partial derivatives for $w$ is

$$\frac{\partial \mathcal{L}_{\text{SAE}}(x; w, z)}{\partial w} = 2z(wz - x) + 2\lambda_2 w. \tag{19}$$

From the KKT condition, we can solve $w = zx/(\lambda_z + z^2)$. Then the loss function without $w$ is

$$\mathcal{L}_{\text{SAE}}(x; z) = \frac{\lambda_2 x^2}{\lambda_2 + z^2} + \lambda_1 h(z).$$

Then consider when $z > 0$, we have

$$\frac{d\mathcal{L}_{\text{SAE}}(x; z)}{dz} = -\frac{2\lambda_2 x^2 z}{(\lambda_2 + z^2)^2} + \lambda_1 h'(z).$$

The $h(z)$ is concave and non-decreasing when $z \geq 0$, so $h'(z)$ is non-increasing and positive.

Consider the function of $m(z) = z/(\lambda_2 + z^2)^2$, we have $m(0) = 0$, $m(\sqrt{\lambda_2/3}) = 9\sqrt{3\lambda_2}/(16\lambda_2^2)$ is a maxima and $m(z) \to 0$ as $z \to \infty$. To obtain two local minima, we need the derivative of $h$ has two intersection points with $2\lambda_2 x^2 z/\{\lambda_1 (\lambda_2 + z^2)^2\}$.

If $h'(z) = 0$ when $z > 0$, then we know there is a local minima at $z = \infty$ since $m(z)$ is decreasing when $z > \sqrt{\lambda_2/3}$. Because $h(|z|)$ is concave, we know that $h(0) \leq h(1)$, leading to another local minima $z = 0$ while $z = 0$ is also the local minima for $m(z)$.

If $\lim_{z \to 0} h'(z) > 0$, then we are sure that $z = 0$ is a local minima for any $x$. Then we just need $9\sqrt{3\lambda_2}x^2/(16\lambda_2) > \lambda_1 h'(\sqrt{\lambda_2/3})$, leading to a interval $[\sqrt{\lambda_2/3}, \sqrt{\lambda_2/3} + c]$ where $\mathcal{L}_{\text{SAE}}(x; z)$ is decreasing with some positive $c$. In other words, there must be a local minima satisfying $z > \sqrt{\lambda_2/3}$.

$\qquad\square$

### E.5. Linear VAE on Composed Linear Dataset

For simplicity, we omit the subindex in $\mathcal{L}_{\text{LVAE}}(\cdot)$ and write its loss function as,

$$\mathcal{L}(\phi, \theta, \gamma) = \int_{\mathcal{X}} \left\{ -\mathbf{E}_{q_\phi(\mathbf{z}|\mathbf{x})} \left[ \log p_\theta(\mathbf{x}|\mathbf{z}) \right] + \mathbf{KL}(q_\phi(\mathbf{z}|\mathbf{x}) \| p(\mathbf{z})) \right\} \omega(d\mathbf{x}), \tag{20}$$

where $q_\phi(\mathbf{z}|\mathbf{x}) = \mathcal{N}(\mathbf{z}|\boldsymbol{\mu}_z, \boldsymbol{\Sigma}_z)$ with $\boldsymbol{\mu}_z \in \mathbb{R}^d$, $p_\theta(\mathbf{x}|\mathbf{z}) = \mathcal{N}(\mathbf{x}|\boldsymbol{\mu}_x, \gamma \mathbf{I})$ with $\boldsymbol{\mu}_x \in \mathbb{R}^\kappa$ and $p(\mathbf{z}) = \mathcal{N}(0, \mathbf{I})$. Here $\boldsymbol{\Sigma}_z \overset{\text{def}}{=} \text{diag}\{\sigma_z^2(\mathbf{x}; \phi)\}$ is a diagonal matrix.

Recall that the decoder here is linear, i.e., $\hat{\mathbf{x}} = \boldsymbol{\mu}_x = \mathbf{W}\mathbf{z}$. Expand the formula in the integral as following

$$\mathcal{L}(\mathbf{x}; \phi, \theta, \gamma) - \mathbf{E}_{q_\phi(\mathbf{z}|\mathbf{x})} \left[ \log p_\theta(\mathbf{x}|\mathbf{z}) \right] + \mathbf{KL}(q_\phi(\mathbf{z}|\mathbf{x}) \| p(\mathbf{z}))$$

$$= -\mathbf{E}_{q_\phi(\mathbf{z}|\mathbf{x})} \left[ -\frac{d}{2} \log(2\pi\gamma) - \frac{\|\mathbf{x} - \boldsymbol{\mu}_x\|^2}{2\gamma} \right] + \frac{1}{2} \left( \text{tr}(\boldsymbol{\Sigma}_z) - \log |\boldsymbol{\Sigma}_z| + \boldsymbol{\mu}_z^\top \boldsymbol{\mu}_z - \kappa \right)$$

$$= \frac{d}{2} \log(2\pi\gamma) + \mathbf{E}_{q_\phi(\mathbf{z}|\mathbf{x})} \left[ \frac{\|\mathbf{x} - \boldsymbol{\mu}_x\|^2}{2\gamma} \right] + \frac{1}{2} \left( \text{tr}(\boldsymbol{\Sigma}_z) - \log |\boldsymbol{\Sigma}_z| + \boldsymbol{\mu}_z^\top \boldsymbol{\mu}_z - \kappa \right)$$

$$= \frac{d}{2} \log(2\pi\gamma) + \mathbf{E}_{q_\phi(\mathbf{z}|\mathbf{x})} \left[ \frac{\|\mathbf{W}\mathbf{z} - \mathbf{x}\|^2}{2\gamma} \right] + \frac{1}{2} \left( \text{tr}(\boldsymbol{\Sigma}_z) - \log |\boldsymbol{\Sigma}_z| + \boldsymbol{\mu}_z^\top \boldsymbol{\mu}_z - \kappa \right)$$

$$= \frac{d}{2} \log(2\pi\gamma) + \frac{1}{2\gamma} \left\{ \text{tr}(\mathbf{W}\boldsymbol{\Sigma}_z \mathbf{W}^\top) + (\mathbf{W}\boldsymbol{\mu}_z - \mathbf{x})^\top (\mathbf{W}\boldsymbol{\mu}_z - \mathbf{x}) \right\}$$

$$+ \frac{1}{2} \left( \text{tr}(\boldsymbol{\Sigma}_z) - \log |\boldsymbol{\Sigma}_z| + \boldsymbol{\mu}_z^\top \boldsymbol{\mu}_z - \kappa \right).$$

The gradient of $\mathcal{L}(\mathbf{x})$ takes the form,

$$\frac{\partial \mathcal{L}(\mathbf{x})}{\partial \boldsymbol{\mu}_z} = \frac{1}{\gamma} \mathbf{W}^\top (\mathbf{W}\boldsymbol{\mu}_z - \mathbf{x}) + \boldsymbol{\mu}_z$$

$$\frac{\partial \mathcal{L}(\mathbf{x})}{\partial \boldsymbol{\Sigma}_z} = \frac{1}{2\gamma} \mathbf{W}^\top \mathbf{W} + \frac{1}{2} (\mathbf{I} - \boldsymbol{\Sigma}_z^{-1})$$

$$\frac{\partial \mathcal{L}(\mathbf{x})}{\partial \mathbf{W}} = \frac{1}{\gamma} \{ \boldsymbol{\Sigma}_z \mathbf{W}^\top + \boldsymbol{\mu}_z \boldsymbol{\mu}_z^\top \mathbf{W}^\top - \boldsymbol{\mu}_z \mathbf{x}^\top \}.$$

Note that $\boldsymbol{\mu}_z$ and $\boldsymbol{\Sigma}_z^{1/2}$ are varying functions of $\mathbf{x}$, then the KKT conditions provide the necessary conditions for an optimal

solution as

$$(\mathbf{W}^\top \mathbf{W} + \gamma \mathbf{I})\boldsymbol{\mu}_z = \mathbf{W}^\top \mathbf{x},$$

$$\boldsymbol{\Sigma}_z^{-1} = \mathbf{W}^\top \mathbf{W}/\gamma + \mathbf{I},$$

$$\int_{\mathcal{X}} (\boldsymbol{\Sigma}_z + \boldsymbol{\mu}_z \boldsymbol{\mu}_z^\top)\mathbf{W}^\top \omega(d\mathbf{x}) = \int_{\mathcal{X}} \boldsymbol{\mu}_z \mathbf{x}^\top \omega(d\mathbf{x}). \tag{21}$$

Here, the first two equations are held for any $\mathbf{x}$ and the last equation is held on integral since $\mathbf{W}$ does not vary with $\mathbf{x}$. For convenience, let $(\gamma \mathbf{I} + \mathbf{W}^\top \mathbf{W})^{-1} = \mathbf{A}$, then $\boldsymbol{\Sigma}_z = \gamma \mathbf{A}$ and $\boldsymbol{\mu}_z = \mathbf{A}\mathbf{W}^\top \mathbf{x}$. Then (21) could be transformed as

$$\int_{\mathcal{X}} (\boldsymbol{\Sigma}_z + \boldsymbol{\mu}_z \boldsymbol{\mu}_z^\top)\mathbf{W}^\top \omega(d\mathbf{x}) = \int_{\mathcal{X}} \boldsymbol{\mu}_z \mathbf{x}^\top \omega(d\mathbf{x}),$$

$$\int_{\mathcal{X}} (\boldsymbol{\Sigma}_z + \boldsymbol{\mu}_z \boldsymbol{\mu}_z^\top)\mathbf{W}^\top \mathbf{W}\mathbf{A}\omega(d\mathbf{x}) = \int_{\mathcal{X}} \boldsymbol{\mu}_z \mathbf{x}^\top \mathbf{W}\mathbf{A}\omega(d\mathbf{x}),$$

$$\int_{\mathcal{X}} (\boldsymbol{\Sigma}_z + \boldsymbol{\mu}_z \boldsymbol{\mu}_z^\top)\mathbf{W}^\top \mathbf{W}\omega(d\mathbf{x}) = \int_{\mathcal{X}} \boldsymbol{\mu}_z \boldsymbol{\mu}_z^\top \omega(d\mathbf{x})(\gamma \mathbf{I} + \mathbf{W}^\top \mathbf{W}),$$

$$\boldsymbol{\Sigma}_z \mathbf{W}^\top \mathbf{W} = \gamma \int_{\mathcal{X}} \boldsymbol{\mu}_z \boldsymbol{\mu}_z^\top \omega(d\mathbf{x}),$$

$$\mathbf{A}\mathbf{W}^\top \mathbf{W} = \int_{\mathcal{X}} \mathbf{A}\mathbf{W}^\top \mathbf{x}\mathbf{x}^\top \mathbf{W}\mathbf{A}\omega(d\mathbf{x}),$$

$$\mathbf{W}^\top \mathbf{W}(\gamma \mathbf{I} + \mathbf{W}^\top \mathbf{W}) = \mathbf{W}^\top \int_{\mathcal{X}} \mathbf{x}\mathbf{x}^\top \omega(d\mathbf{x})\mathbf{W},$$

$$\mathbf{W}^\top (\gamma \mathbf{I} + \mathbf{W}\mathbf{W}^\top)\mathbf{W} = \mathbf{W}^\top \int_{\mathcal{X}} \mathbf{x}\mathbf{x}^\top \omega(d\mathbf{x})\mathbf{W}. \tag{22}$$

Denote the singular value decomposition of $\mathbf{W}$ as $\mathbf{W} = \mathbf{U}\mathbf{S}\mathbf{V}^\top$ with $\mathbf{U} \in \mathbb{R}^{d \times \kappa}$, $\mathbf{S} \in \mathbb{R}^{\kappa \times \kappa}$ being a diagonal matrix, and $\mathbf{V} \in \mathbb{R}^{\kappa \times \kappa}$. Then (22) can be simplified as

$$\mathbf{V}\mathbf{S}\mathbf{U}^\top (\gamma \mathbf{I} + \mathbf{U}\mathbf{S}^2\mathbf{U}^\top)\mathbf{U}\mathbf{S}\mathbf{V}^\top = \mathbf{V}\mathbf{S}\mathbf{U}^\top \int_{\mathcal{X}} \mathbf{x}\mathbf{x}^\top \omega(d\mathbf{x})\mathbf{U}\mathbf{S}\mathbf{V}^\top$$

Note that we have $\mathbf{V}^\top \mathbf{V} = \mathbf{I}_\kappa$, and $\mathbf{U}^\top \mathbf{U} = \mathbf{I}_d$, then we have

$$\gamma \mathbf{S}^2 + \mathbf{S}^4 = \mathbf{S}\mathbf{U}^\top \int_{\mathcal{X}} \mathbf{x}\mathbf{x}^\top \omega(d\mathbf{x})\mathbf{U}\mathbf{S}. \tag{23}$$

Notice the left side is a diagonal matrix, thus $\mathbf{U}^\top \int_{\mathcal{X}} \mathbf{x}\mathbf{x}^\top \omega(d\mathbf{x})\mathbf{U} \triangleq \boldsymbol{\Sigma}_x^2$ should also be diagonal. Since $\mathbf{x}$ is lying in a composed linear space, then the rank of $\int \mathbf{x}\mathbf{x}^\top \omega(d\mathbf{x})$ must be the dimension of smallest linear space covering all samples.

Denote the dimension by $r$, and assume the first $r$ diagonal element in $\boldsymbol{\Sigma}_x^2$ is nonzero. Finally, we obtain that $\mathbf{S}_{ii} = \sqrt{\sigma_{xi}^2 - \gamma}$ when $i \leq r$ and $\mathbf{S}_{ii} = 0$ when $i > r$, where recall $\mathbf{S}_{ii}$ is the $i$-th diagonal element in $\mathbf{S}$ and $\sigma_{xi}^2$ is the short of $\boldsymbol{\Sigma}_{x,ii}^2$. Now the interim matrix $\mathbf{A}$ is $\mathbf{V}\text{diag}\{\sigma_{x1}^{-2}, \ldots, \sigma_{xr}^{-2}, \gamma^{-1}, \ldots, \gamma^{-1}\}\mathbf{V}^\top$, and $\boldsymbol{\Sigma}_z = \mathbf{V}\text{diag}\{\gamma \sigma_{x1}^{-2}, \ldots, \gamma \sigma_{xr}^{-2}, 1, \ldots, 1\}\mathbf{V}^\top$. So the number of active dimensions is $r$.

To compute the loss function, we also need

$$\int_{\mathcal{X}} \|\mathbf{W}\boldsymbol{\mu}_z - \mathbf{x}\|_2^2 \omega(d\mathbf{x}) = \int_{\mathcal{X}} \|(\mathbf{W}\mathbf{A}\mathbf{W}^\top - \mathbf{I})\mathbf{x}\|_2^2 \omega(d\mathbf{x})$$

$$= \int_{\mathcal{X}} \|\mathbf{U}\text{diag}\{-\gamma \sigma_{x1}^{-2}, \ldots, -\gamma \sigma_{xr}^{-2}, -1, \ldots, -1\}\mathbf{U}^\top \mathbf{x}\|_2^2 \omega(d\mathbf{x})$$

$$= \text{tr}\left(\text{diag}\{\gamma^2 \sigma_{x1}^{-2}, \ldots, \gamma^2 \sigma_{xr}^{-2}, 1, \ldots, 1\}\mathbf{U}^\top \int_{\mathcal{X}} \mathbf{x}\mathbf{x}^\top \omega(d\mathbf{x})\mathbf{U}\right)$$

$$= \gamma^2 \sum_{i \leq r} \sigma_{xi}^{-2},$$

and

$$\int_{\mathcal{X}} \|\boldsymbol{\mu}_z\|_2^2 \omega(\mathrm{d}\mathbf{x}) = \int_{\mathcal{X}} \|\mathbf{A}\mathbf{W}^\top \mathbf{x}\|_2^2 \omega(\mathrm{d}\mathbf{x}) = \int_{\mathcal{X}} \mathbf{x}^\top \mathbf{W}\mathbf{A}^2\mathbf{W}^\top \mathbf{x}\omega(\mathrm{d}\mathbf{x})$$

$$= \int_{\mathcal{X}} \mathbf{x}^\top \mathbf{U}\mathrm{diag}\{(\sigma_{x1}^2 - \gamma)/\sigma_{x1}^4, \ldots, (\sigma_{xr}^2 - \gamma)/\sigma_{xr}^4, 0, \ldots, 0\}\mathbf{U}^\top \mathbf{x}\omega(\mathrm{d}\mathbf{x})$$

$$= \mathrm{tr}\left(\mathrm{diag}\{(\sigma_{x1}^2 - \gamma)/\sigma_{x1}^4, \ldots, (\sigma_{xr}^2 - \gamma)/\sigma_{xr}^4, 0, \ldots, 0\}\mathbf{U}^\top \int_{\mathcal{X}} \mathbf{x}\mathbf{x}^\top \omega(\mathrm{d}\mathbf{x})\mathbf{U}\right)$$

$$= \sum_{i \leq r}(\sigma_{xi}^2 - \gamma)/\sigma_{xi}^2.$$

With these results, the minimum of energy is

$$\mathcal{L}_{(\phi_\gamma^*, \theta_\gamma^*, \gamma)} = \int_{\mathcal{X}} \frac{d}{2}\log(2\pi\gamma) + \frac{1}{2\gamma}\left\{\gamma\sum_{i \leq r}(\sigma_{xi}^2 - \gamma)/\sigma_{xi}^2 + \|\mathbf{W}\boldsymbol{\mu}_z - \mathbf{x}\|_2^2\right\}$$

$$+ \frac{1}{2}\left\{\sum_{i \leq r}\gamma/\sigma_{xi}^2 + \kappa - r - \sum_{i \leq r}(\log\gamma - \log\sigma_{xi}^2) + \|\boldsymbol{\mu}_z\|_2^2 - \kappa\right\}\omega(\mathrm{d}\mathbf{x})$$

$$= \frac{d}{2}\log(2\pi\gamma) + \frac{1}{2}\sum_{i \leq r}(\sigma_{xi}^2 - \gamma)/\sigma_{xi}^2 + \frac{1}{2}\gamma\sum_{i \leq r}\sigma_{xi}^{-2}$$

$$+ \frac{1}{2}\{\gamma\sum_{i \leq r}\sigma_{xi}^{-2} - r - r\log\gamma + \sum_{i \leq r}\log\sigma_{xi}^2 + r - \gamma\sum_{i \leq r}\sigma_{xi}^{-2}\}$$

$$= \frac{d}{2}\log(2\pi\gamma) - \frac{r}{2}\log\gamma + \frac{1}{2}\sum_{i \leq r}\log\sigma_{xi}^2 + \frac{1}{2}r. \tag{24}$$

As $\gamma \to 0$, the rate of minimum energy is $(d - r)/2\log\gamma + O(1)$.

$\square$

