# OpenReview forum: "Sparse Autoencoders, Again?"
_ICML.cc/2025/Conference — ICML 2025 poster_

### Official Review · Reviewer_75rJ · 2025-03-09

**Overall Recommendation:** 3

**Summary:**

The paper proposes sparse variational autoencoders as an analog to sparse autoencoders by adding a sample wise sparsity mask.
The authors then consider low dimentional data and show that only this low dimensions are active for optimal parameters. They then compare this model on multiple real world data sets to validate that it can outperform other sparse autoencoder benchmarks.

**Claims And Evidence:**

The claims are supported by evidence however I have some concerns about the theory part, see theoretical claims below.

After the rebuttal the concerns have been addressed

**Essential References Not Discussed:**

None

**Experimental Designs Or Analyses:**

I did not check the experiments in detail.

**Methods And Evaluation Criteria:**

The benchmarks seems sensible, it should be noted that the only comparison made is against other sparse autoencoder models.

**Other Comments Or Suggestions:**

See questions below

**Other Strengths And Weaknesses:**

(+)The idea for this paper is well motivated and the paper is well structured
(+) The novel ideas in this paper warrant exploration and I believe this to be a good direction for research.
(-) The overall contribution in this paper feels a bit shallow in the sense that the theoretical claims are quite surface level and the experiments are still on simpler (if real world) data and compared against other sparse autoencoder models only. There is not one particular thing wrong, but it seems to lack a strong point that really has an impact. I would be happy to have my mind changed on this by the authors rebuttal
(-) The theoretical contribution is not very clear both the claims themselves as well as the proofs (see questions for below)

**Questions For Authors:**

- I am a bit confused about definition 4.1. Manifolds of dimension $r$ that are diffeomorphic to $\mathbb{R}^r$, are in particular diffeomorphic to an open ball in $\mathbb{R}^r$ (as $\mathbb{R}^r$ is diffeomorphic to a ball). This seems very restrictive, as it means that up to differomorphic equivalence only open balls are considered. In particular this excludes any compact manifold (e.g. sphere) as well as manifolds with boundary. From an applied perspective this restriction seems strange, could the authors elaborate on why it is needed and whether it can be relaxed?
- Regarding definition 4.3, why does $\sigma_z(x;\phi_\gamma) = \Theta(1)$ correspond to inactive dimensions, in principle we could still have $1 - \sigma_z(x;\phi_\gamma) = \Theta(1)$. Why do we not need that $ \sigma_z(x;\phi_\gamma)  \approx 1$?
- The notation is not quite clear: in (4) $\sigma_z[x;\phi]$ is defined, is this different from $\sigma_z(x;\phi) defined in (2) ? For example in Definition 4.3 the () brackets are used.
- For theorem 4.5 you need $\sum_i r_i \leq \kappa$. Do you have any intuition what happens if this is not the case?
- Why is 4.6 called a Corollary, how does it follow from Theorem 4.5 (it seemed that rather than being deduced it was proven in a similar manner)
- The first paragraph in F.1 states that you first analyze and then construct a solution, but afterwards you seem to construct a solution and then analyze.

**Relation To Broader Scientific Literature:**

The results further the understanding of sparse autoencoders by considering variational autoencoders. It can also be of more general interest to representation and interpretability as the learned structure is quite explicit.

**Theoretical Claims:**

I went through the proofs in appendix F, they seemed correct, comments are in questions.

---

> ### Author Rebuttal · Authors · 2025-03-31
>
> Thanks for the comprehensive review of our work, particularly with respect to checking proof details and pointing out multiple valuable corrections.
>
> **Comment:**
> *The benchmarks seems sensible ... but the only comparison made is against other sparse autoencoder models.*
>
> **Response:**
> Actually, we also compare against *diffusion* in Section 5.3 which is definitively not an SAE.  Additionally, we compare extensively against vanilla VAEs, which have not historically been considered as SAEs per se.  Still, the predominant focus of our work is indeed on improving upon SAE models.
>
>
> **Comment:**
> *I did not check the experiments in detail ... Went through the proofs in appendix F...*
>
> **Response:**
> Thanks for checking the proofs, *this is a great help*.  The diversity and complementary nature of our experiments also serve as a key part of our contribution.
>
> **Comment:**
> *(+)The idea for this paper is well motivated and the paper is well structured (+) The novel ideas in this paper warrant exploration and I believe this to be a good direction for research.*
>
> **Response:**
> We appreciate acknowledgement of the novelty of our work, and indeed no prior paper has repurposed and analyzed VAE models in the way that we have.
>
> **Comment:**
> *(-) The overall contribution in this paper feels a bit shallow ... comparisons against other sparse autoencoder models only ...*
>
> **Response:**
> We emphasize that VAEs represent one of the highest cited, influential ML frameworks of the last 10 years, and yet they have never been previously applied or analyzed in the context of SAE tasks as we have done.  Nor have the specific attributes of adaptive sparsity, local minima smoothing, and guaranteed maximally sparse global optima been cohesively addressed in the context of widely-used SAE models.  Still, as a contribution to an ML conference we do not match the mathematical depth or generality of a pure theoretical contribution on manifold geometry, etc.  Instead, we produce a brand new VAE use case, *that requires no additional model complexity*, and elucidate unique attributes shared by no prior approaches.  We also offer extensive supporting experimentation, including comparisons with diffusion models beyond SAEs alone.
>
>
> **Comment:**
> *(-) The theoretical contributions need clarification ...*
>
> **Response:**
> Please see our specific responses elsewhere.  However, broadly speaking, our results establish that the proposed VAEase energy function uniquely and simultaneously achieves two desirable properties: (i)  The global minima exactly reflect union-of-manifold structure in data, and (ii) the loss exhibits a selective smoothing mechanism capable of reducing local minimizers.  *No prior SAE or VAE approach achieves both*; provably so in the sense described in Section 4 for the wide classes we consider as baselines.
>
>
> **Comment:**
> *Question about Definition 4.1 and and whether stated conditions can be relaxed, etc.*
>
> **Response:**
> Great suggestion.  The issue here relates to an inadvertent typo in our submission, where we used $\mapsto$ instead of $\rightarrow$ in Definition 4.1.  Crucially, we do *not* require the image set in $\mathbb{R}^r$ to be an open ball. This implies: (i) the manifold's image can be an arbitrary topology as long as a global diffeomorphism exists; (ii) manifolds with a boundary remain valid objects of study as the definition focuses on the dimensionality of image set rather than imposing openness constraints.
>
>
>
> **Comment:**
> *Regarding Definition 4.3, do we really need $\sigma_z(x;\phi_\gamma) = \Theta (1)$?*
>
> **Response:**
> In the proof from Section F.1 (Line 1003), the correct asymptotic requirement should be $\sigma_z^2(x;\phi_\gamma) = O(\gamma)$ for *active dims* or $1 - \sigma_z^2(x;\phi_\gamma) = O(\gamma)$ for *inactive dims* to eliminate constants in the inequality. We will clarify the revision accordingly; thanks for pointing out this oversight in our original draft.
>
>
> **Comment:**
> *The notation is not quite clear ... specifically brackets vs parentheses in equations (2) and (4).*
>
> **Response:**
> We adopt the convention (shared by some ML works) that parentheses and brackets are interchangeable, and are chosen purely for making nested cases more readable, i.e., to avoid multiple parentheses or brackets in a row that can be visually difficult to parse.  But this is obviously a minor stylistic choice that can be changed if the reviewer prefers.
>
>
> **Comment:**
> *Why is 4.6 called a Corollary ...*
>
> **Response:**
> Corollary 4.6 involves analysis closely related to Theorem 4.5, and the final conclusions are complementary.  Still, we can easily change this designation if preferable.
>
> **Comment:**
> *The first paragraph in F.1 is not clear, etc...*
>
> **Response:**
> Great catch.  Indeed, the original text was backwards; we will correct it to state that we first construct a feasible solution and then analyze to form a necessary condition for optimality.

---

> > ### Comment · Reviewer_75rJ · 2025-04-06
> >
> > I thank the authors for addressing my concerns I have updated my review accordingly.

---

> > > ### Author Response · Authors · 2025-04-08
> > >
> > > Thanks again for evaluating technical aspects of our paper in particular and for engaging with our rebuttal.

---

### Official Review · Reviewer_WZdZ · 2025-03-10

**Overall Recommendation:** 4

**Summary:**

After a discussion of the strengths and weaknesses of variational autoencoders (VAEs) and sparse autoencoder (SAEs), this paper proposes an adaptation of the VAE architecture to enhance sparsity and target the interpretability objective that motivates SAEs. The idea is conceptually simple: use the learned variance network $\sigma_z$ as a mask to select the active latent dimensions of z by element-wise multiplying the latent representations z by 1 - $\sigma_z$. The authors theoretically show that this adjustment allows to benefit from the local minima smoothing properties of VAEs while mimicking the behavior of SAEs. Then, the authors present empirical evidence supporting the effectiveness of the proposed VAEase architecture. In particular, they compare it to a vanilla VAE, and SAEs in which sparsity is enforced by L1 regularization, log and top-k activations. The methods are evaluated in a toy setting in which data are generated in a small number of low-dimensional manifolds and then embedded into a space with larger dimensionality. Additionally, the auto-encoding methods are evaluated on the reconstruction of image data (MNIST and Fashion-MNIST) and of the activations of a Pythia language model. Finally, the paper presents a comparison of VAEase to diffusion models on the reconstruction of pseudo-MNIST data generated using a GAN.

**Claims And Evidence:**

The authors claim that the VAEase architecture is able to achieve a better sparsity-reconstruction trade-off than SAEs and vanilla VAEs. This claim is adequately supported by theoretical and empirical results.

**Essential References Not Discussed:**

None.

**Experimental Designs Or Analyses:**

The experimental analyses in the toy setting and on image and language model data are sound. However, given that one of the main applications of SAEs is in interpreting language model activations, additional empirical validation of language model data (e.g., on a more recent, larger model) would make the paper stronger.

**Methods And Evaluation Criteria:**

The experimental setup in both the toy setting and on image and language model data is reasonable. The evaluation metrics for sparsity and reconstruction quality are appropriate.

**Other Comments Or Suggestions:**

None.

**Other Strengths And Weaknesses:**

None.

**Questions For Authors:**

- What does the reconstruction error/sparsity trade-off for language model data look like? A visualization similar to Figure 3 (possibly in the appendix) would be appreciated.

**Relation To Broader Scientific Literature:**

The contributions of the paper connect with the literature about VAEs and with recent works proposing the application of SAEs to interpret language model activations. This is adequately discussed by the authors in Section 2.

**Theoretical Claims:**

I did not find any issues in the proofs of the theoretical claims. However, it’s possible that I might have missed something.

---

> ### Author Rebuttal · Authors · 2025-03-31
>
> Thanks for the quite comprehensive and accurate summary of our work.  Indeed this description captures the essence of what we hoped to convey.
>
> **Comment:**
> *...additional empirical validation of language model data (e.g., on a more recent, larger model) would make the paper stronger.*
>
> **Response:**
> We agree that further investigation into LLM applications of our work is desirable.  However, because we are not domain specialists w.r.t. LLM interpretability in particular (which involves a number of qualitative investigations), we have deferred this application-specific dimension to future work.  We also remark that prior SAE papers devoted to LLMs have proposed the same metrics that we adopt herein for quantification purposes.
>
> That being said, as our emphasis is on diversity of application, especially very recent ones, we have added an additional LLM-driven experiment whereby the downstream goal is to generate hypotheses regarding relationships between text data (e.g., headlines) and a target variables (e.g., clicks); please see https://arxiv.org/pdf/2502.04382.  While we defer details to this prior paper (no affiliation to us), the key optimization task involves learning sparse representations of text embeddings from Yelp reviews.  Results are shown in the new table below, where we observe that VAEase again requires the fewest average active dimensions (AAD) even while maintaining the best reconstruction error (RE).  This further expands the diversity of scenarios whereby VAEase outperforms existing alternatives.  Thanks for the suggestion.
>
>
> | Model     | RE     | AAD         |
> |-----------|--------|------------|
> | SAE-$\ell_1$    | 0.2236 | 61.92 |
> | SAE-$\log$   | 0.2298 | 55.97 |
> | SAE-$T_k$  | 0.2168 | 30.00   |
> | VAE       | 0.2774 | 90.00   |
> | VAEase    | **0.1869** | **16.74** |
>
>
> **Comment:**
> *What does the reconstruction error/sparsity trade-off for language model data look like? A visualization similar to Figure 3 (possibly in the appendix) would be appreciated.*
>
> **Response:**
> Good question.  While Openreview does not allow us to upload new figures, we have run this experiment and the trend closely matches that from Figure 3.  This further solidifies the stable advantage of VAEase across quite distinct domains.  And such a figure can easily be added to a revised appendix as the reviewer suggests.

---

### Official Review · Reviewer_LJBm · 2025-03-12

**Overall Recommendation:** 4

**Summary:**

The paper introduces a method to explicitly model sparsity in variational autoencoders (VAEs) that leverages a simple (parameter-free) transformation of the latents before decoding. The method is introduced by intuitive construction and supported by theoretical arguments and results. Experimental comparisons show the effectiveness of the method in learning sparse representations of data.

## Update after rebuttal
After clarifications from the authors regarding my criticism of the active dimension removal experiment and additional experiments provided in response to another reviewer, I updated my score to 4 (Accept) to better represent my evaluation of the paper.

**Claims And Evidence:**

The claims are overall reasonable and the evidence provided is mostly convincing.

**Essential References Not Discussed:**

Not applicable.

**Experimental Designs Or Analyses:**

I think the experiments conducted are adequate overall, except for the few points below.

As far as I understand, the masking of active dimensions in the experiment illustrated in Fig. 3 follows the partitioning criteria described in lines 367--384 ("throughout this paper" is stated therein). Would it not be fairer, when viewing this against reconstruction error, to mask dimensions in order of least to most disturbing w.r.t. the reconstruction error? In other words, sequentially remove the next active dimension that would perturb the average reconstruction loss the least (maybe different dimensions are picked per model).

In lines 420--423: This sentence in the analysis is faulty in my view, since their own method estimates $r \approx 15$ on Pseudo-MNIST. Since the setup upper bounds the ground-truth $r$, a simple (but very laborious) test could follow this setup from 16 dimension down to one dimension, stopping whenever the whole setup degenerates.

**Methods And Evaluation Criteria:**

The experimental setup is valid and the proposed method is sound for the problem of interest.

**Other Comments Or Suggestions:**

Some typos/mistakes:
- l. 051: hyperparaemters -> hyperparameters
- l. 196: DGMs? Perhaps DMs is meant here?
- l. 256: $\mapsto$ should be $\rightarrow$ (`\rightarrow`)
- l. 235-236: citation uses parenthesis
- l. 318: "large" seems to no belong in the sentence
- l. 391: "projection" is a somewhat overloaded term, but has a precise meaning as a linear function. Perhaps use "transformation" or "mapping"?
- l. 847: "encoders" is mentioned twice
- l. 854: wieghts -> weights
- l. 980: isn't $\mathcal{X}'$ meant in the integral?

Minor comments:
- The title of section C.1 struck me as quite colloquial
- F.1: $\mathbf{e}$ is often used as the canonical orthonormal basis vectors, perhaps a better name would be $\mathbf{1}$?
- l. 998: Could you indent the expression better to be aligned inside the integral?

**Other Strengths And Weaknesses:**

**Strengths**

The paper is very well written and organised, which made it very easy to read. The idea pursued is simple, well-motivated, and with a straightforward implementation, leading to a high potential of being used by those interested in sparse autoencoders.

**Weaknesses**

A worthy mention is perhaps the applicability of sparse autoencoders in general, coupled with the size of the change proposed to VAEs. While properly mathematically substantiated and experimentally verified, the main contribution is still arguably quite straightforward and that could be viewed as its weakness.

**Questions For Authors:**

None at this point.

**Relation To Broader Scientific Literature:**

The method introduced enable variational autoencoders to be used effectively as sparse autoencoders. This naturally builds upon both sparse autoencoders and variational autoencoders, the former now experiencing increased relevance due to their use in understanding the "inner" representations used by large language models.

**Theoretical Claims:**

I checked the correctness of the proof for Theorem 4.5. Despite minor writing/formatting issues and convenient assumptions which I'm not sure about regarding applicability in pratical cenarios (Lipschitz continuity), I have no issues to mention.

---

> ### Author Rebuttal · Authors · 2025-03-31
>
> We are appreciative of the detailed, constructive comments, and for pointing out the high potential of our work being used by those interested in sparse autoencoders (which is our intended audience). We address main reviewer points in turn below.
>
> **Comment:**
> *As far as I understand, the masking of active dimensions in the experiment illustrated in Fig. 3 follows the partitioning criteria described in lines 367--384 ("throughout this paper" is stated therein). Would it not be fairer, when viewing this against reconstruction error, to mask dimensions in order of least to most disturbing w.r.t. the reconstruction error? In other words, sequentially remove the next active dimension that would perturb the average reconstruction loss the least (maybe different dimensions are picked per model).*
>
> **Response:**
> Actually, the shape of Figure 3 does not appreciably change when enacting the reviewer's suggestion.  This is because the magnitude of $\mu_z$ values for SAE models, or $\sigma_z$ values for VAE models, already closely reflect their importance w.r.t. reconstruction error by design.  Moreover, using these magnitudes better reflects practical use cases, whereby we would ideally like to assess active dimensions without having to compute $O(\kappa^2)$ separate reconstruction errors.
>
>
> **Comment:**
> *In lines 420--423: This sentence in the analysis is faulty in my view, since their own method estimates $r \approx 15$ on Pseudo-MNIST. Since the setup upper bounds the ground-truth $r$, a simple (but very laborious) test could follow this setup from 16 dimension down to one dimension, stopping whenever the whole setup degenerates.*
>
> **Response:**
> Our intended point on Lines 420-423 and the surrounding text is the following:  By design of the experiment, we know that 16 is a strict upper bound on the ground-truth manifold dimension.  Hence any method that produces an estimate above 16 must be incorrect.  In particular, both diffusion approaches in Table 4 produce estimates vastly exceeding 16.  Meanwhile our VAEase approach produces an estimate below 16 despite no knowledge of the generative process.  Hence the VAEase estimate is at least plausible, even while conceding that the true manifold dimension might still be a bit smaller (which would also be even further from the diffusion estimates).  Of course intuition suggests it can't be dramatically smaller, as visual inspection of human-written MNIST digits indicates multiple non-trivial degrees of freedom.
>
> Overall though, our main objective here is to show the superiority of VAEase relative to diffusion, not exactly pinpoint any exact manifold dimensions associated with MNIST.  The reviewer's suggestion would indeed help with the latter, but this is computationally intensive and involves subjectively deciding when GAN samples deviate too far from MNIST. Still, the reviewer's comment is well-noted, and we will adjust the writing to ensure our desired points are more clearly conveyed.
>
>
> **Comment:**
> *I reviewed most of the supplementary material. It was surprising to me how the quality of the writing is considerably worse in the appendix sections compared to the main paper.*
>
> **Response:**
> Admittedly in the rush to submit, more attention was devoted to polishing the main text, which the reviewer agrees is well written and organized.  For the revision we can definitely remedy this imbalance.
>
>
> **Comment:**
> *A worthy mention is perhaps the applicability of sparse autoencoders in general, coupled with the size of the change proposed to VAEs. While properly mathematically substantiated and experimentally verified, the main contribution is still arguably quite straightforward and that could be viewed as its weakness.*
>
> **Response:**
> With regard to SAEs, we believe their relevance is increasing along multiple fronts.  For example, as we discuss in Section 5 and elsewhere, there is rapidly growing usage w.r.t. learning interpretable representations of LLM activation layers. Moreover, other representative SAE examples spanning vision and language are frequently appearing in the literature, e.g., as in https://arxiv.org/pdf/2502.04382, https://arxiv.org/pdf/2502.06755v1, and https://arxiv.org/pdf/2410.06981.  And to better reflect this trend, we have added additional related experiments; please see our rebuttal response to Reviewer WZdZ for details.
>
> Secondly, although our VAEase proposal is admittedly simple, we view this as a fundamental strength as it is easy for anyone to apply, portending larger downstream impact.  Moreover, the underlying novelty is evidenced by 10 years of extensive VAE usage without prior work ever uncovering, let alone rigorously analyzing, the highly-effective modification we have proposed.
>
>
> **Comment:**
> *Some typos/mistakes ... Minor comments*
>
> **Response:**
> Thanks for pointing these out, it is extremely helpful for improving the revision.  We will definitely correct each of them.

---

> > ### Comment · Reviewer_LJBm · 2025-04-07
> >
> > **RE: Removal of active dimensions based on reconstruction error**
> >
> > I see. This is overall great, but I have one more comment:
> >
> > > Moreover, using these magnitudes better reflects practical use cases, whereby we would ideally like to assess active dimensions without having to compute separate $O(\kappa^2)$ reconstruction errors.
> >
> > I disagree that this is an argument in favour of the evaluation scheme (or against, for that matter). I believe the reasoning should be to effectively evaluate your method. Whether this is useful for practice later is a separate (but also important) concern.
> >
> > **RE: Lines 420--432**
> >
> > Thanks for the clarification!
> >
> > **RE: Weakness**
> >
> > I agree and appreciate the additional experiment. I am increasing my score after a second look into the authors' rebuttal and  additional experiments.

---

> > > ### Author Response · Authors · 2025-04-08
> > >
> > > Thanks for closely considering our rebuttal points and providing further constructive feedback.

---

### Official Review · Reviewer_JzHL · 2025-03-12

**Overall Recommendation:** 3

**Summary:**

This paper addresses the limitations of traditional Sparse Autoencoders (SAEs) and Variational Autoencoders (VAEs) in sparse representation learning, particularly their inability to adaptively adjust sparsity patterns and sensitivity to hyperparameters. The authors propose a novel model called VAEase, which combines the strengths of both SAEs and VAEs by introducing a new mechanism that dynamically adjusts sparsity based on input samples without requiring hyperparameter tuning. VAEase achieves this by modifying the VAE framework to use encoder variance as a gating mechanism, allowing it to selectively activate or deactivate latent dimensions for each input. Empirical evaluations on synthetic and real-world datasets demonstrate that VAEase outperforms existing SAE and VAE models in accurately estimating underlying manifold dimensions and producing sparser representations while maintaining low reconstruction error. Overall, VAEase provides a more flexible and efficient approach to sparse autoencoding tasks.

**Claims And Evidence:**

There is clear and convincing evidence of the claims made in the submission.

**Essential References Not Discussed:**

None

**Experimental Designs Or Analyses:**

The benchmark dataset and evaluation metrics in this paper are not popular in the field of autoencoder research.

**Methods And Evaluation Criteria:**

The proposed method make sense for the problem， but I do not find the advantages compared with traditional VAE.

The benchmark dataset and evaluation metrics in this paper are not popular in the field of autoencoder research.

**Other Comments Or Suggestions:**

None

**Other Strengths And Weaknesses:**

Strengths：

1. The proposed VAEase model introduces a novel yet straightforward modification to the traditional VAE framework by leveraging encoder variance as a gating mechanism. This simple change effectively enables adaptive sparsity without complicating the overall model architecture.

2. The paper provides comprehensive theoretical proofs that support the effectiveness of VAEase. The authors rigorously demonstrate that VAEase can recover underlying manifold structures and prove that it has fewer local minima compared to traditional SAEs, thereby solidifying the theoretical foundation of the proposed method.

3. The experiments are well-designed and demonstrate clear improvements of VAEase over existing methods. The model consistently outperforms SAEs, VAEs, and diffusion models across various synthetic and real-world datasets, validating its ability to achieve sparser representations with lower reconstruction errors.

4.  The authors test VAEase on diverse datasets, including linear and nonlinear synthetic data, image datasets (MNIST and FashionMNIST), and high-dimensional language model activations. The consistent performance across these datasets highlights the robustness and generalizability of the proposed model.

5.  The paper is well-written, with clear explanations of the model design, theoretical analyses, and experimental results. The authors effectively communicate complex ideas in an accessible manner, making it easy for readers to understand the innovations and contributions of their work.

Weaknesses:

1.  The experiments primarily focus on synthetic datasets, MNIST, FashionMNIST, and intermediate activations of language models. While these datasets are useful for initial validation, they lack the complexity and diversity of real-world image datasets like ImageNet or COCO. Testing on such mainstream datasets would provide stronger evidence of the model's practical applicability and robustness.

2.  Although the paper claims improvements over traditional VAEs, the reported gains might not be substantial enough to justify the additional complexity introduced by the VAEase model. In some cases, the differences in performance metrics (e.g., reconstruction error, sparsity) may appear marginal, raising questions about the model's overall advantage.

3.  The paper emphasizes the theoretical and experimental aspects of VAEase but lacks a clear discussion on its practical applications. The benefits of adaptive sparsity and reduced reconstruction error are not directly linked to specific use cases, making it difficult to assess the model's real-world impact and potential adoption in industry or other research areas.

4.  While the paper provides extensive theoretical proofs, the innovation in the model design might not be immediately apparent to readers. The authors could benefit from a more concise summary of the key innovations and their implications, rather than focusing solely on detailed proofs. This would help highlight the model's novelty and contributions more clearly.

**Questions For Authors:**

Please refer to the above weaknesses.


## update after rebuttal

Based on the author's responses and the comments from other reviewers, I still maintain my score of weak accept and lean towards accepting the paper. Although I am not an expert of sparse autoencoder, I still believe this work is very meaningful.

**Relation To Broader Scientific Literature:**

None

**Theoretical Claims:**

The proofs in the article are very detailed. I didn’t fully understand them, but I think they are valid and correct.

---

> ### Author Rebuttal · Authors · 2025-03-31
>
> Thanks for acknowledging the many positive aspects of our work, including the novel design, comprehensive proofs, solid theoretical foundation, well-designed experiments, and the robustness and generalizability of our proposed model.  We also appreciate the reviewer's statement that there is clear and convincing evidence of the claims made in our submission.
>
> **Comment:**
> *The proposed method make sense for the problem, but I do not find the advantages compared with traditional VAE.*
>
> **Response:**
> A traditional VAE model cannot readily learn adaptive sparse solutions (meaning solutions whereby the locations of informative latent dimensions vary from sample to sample).  In fact, we rigorously prove this VAE limitation via Corollary 4.6, and verify it empirically; see for example Table 2, where the VAE is incapable of learning the correct manifold structure *which requires adaptive sparsity to do so*.
>
> **Comment:**
> *The benchmark dataset and evaluation metrics in this paper are not popular in the field of autoencoder research.*
>
> **Response:**
> SAE models are commonly applied to a wide variety of different tasks, such that we are not aware that there is any particular most popular benchmark.  That being said, arguably the most high-profile recent application is to the analysis of LLM activation layers, and we have included such experimentation in our paper.  We have also added another new application example to the rebuttal; please see our response to Reviewer WZdZ. Moreover, the metrics we adopt closely mirror prior work as well.
>
> **Comment:**
> *1. ... Testing on more diverse image datasets like ImageNet or COCO.*
>
> **Response:**
> While we agree further testing can always be valuable, our focus is not specifically on computer vision tasks, which is why our submission also includes multiple non-image examples.  Moreover, we do not commonly see SAE models applied to ImageNet or COCO.  Still, application to broader image classes is a useful suggestion for future work.
>
> **Comment:**
> *2. Although the paper claims improvements over traditional VAEs, reported gains might not be substantial enough to justify the additional complexity introduced by the VAEase model ...*
>
>
> **Response:**
> We respectfully disagree with this point for multiple key reasons.  First and most importantly, *our proposed VAEase approach does not introduce additional complexity over a regular VAE*.  Instead, it merely reuses the exact same encoder variances as a novel gating mechanism; hence no further justifications are needed with regard to complexity.  And secondly, for a fixed reconstruction error, VAEase does in fact produce much greater sparsity across a wide range of tasks as desired.  Of course the reported reconstruction errors are often similar, but this is specifically part of the experimental design to facilitate meaningful sparsity comparisons.  Along these lines, we also note that under listed strengths, the reviewer specifically mentioned that our experiments are "well-designed and demonstrate clear improvements of VAEase over existing methods."
>
> **Comment:**
> *3. The paper emphasizes the theoretical and experimental aspects of VAEase but lacks a clear discussion on its practical applications ... benefits of adaptive sparsity and reduced reconstruction error are not directly linked to specific use cases.*
>
> **Response:**
> Given the vast literature involving SAE models, we largely defer treatment of particular practical applications to prior work.  That being said, we do link the benefits of adaptive sparsity and reduced reconstruction errors to the specific use case of LLM activation interpretability.  In particular, to complement more subjective/qualitative analysis of SAE-learned representations, we follow common practice in the LLM literature that advocates specifically for comparing sparsity levels at a shared reconstruction error, with greater sparsity (meaning adaptive sparsity from sample to sample as our VAEase achieves) being associated with better interpretability.  Even so, we are happy to include further references to this effect (although space in our submission is tight, so there is quite limited room for much additional discussion).
>
>
> **Comment:**
> *4. ...The authors could benefit from a more concise summary of the key innovations and their implications, rather than focusing solely on detailed proofs.*
>
> **Response:**
> As an important clarification, our work does not focus solely on detailed proofs.  In fact, all proofs are exclusively deferred to the very end of our paper deep in the appendices, and certainly not treated as a central focus.  And for reference, we concisely list our central innovations and their consequences on the bulleted paragraphs beginning on Line 60 as well as pointers to other sections therein.

---

> > ### Comment · Reviewer_JzHL · 2025-04-07
> >
> > Thanks for the rebuttal that addressed most of my concerns. I will keep my score and tend to accept this paper.

---

> > > ### Author Response · Authors · 2025-04-08
> > >
> > > We appreciate the reviewer's acknowledgement that our rebuttal addressed most concerns.  And if any lingering issue remains, we are happy to address before the discussion periods ends.

---

### Decision · Program_Chairs · 2025-05-01

**Decision:**

Accept (poster)

**Comment:**

The paper proposes a novel model that combines the strengths of both sparse and variational autoencoders by introducing a mechanism that adjusts sparsity based on input samples without requiring hyperparameter tuning. This is a solid paper and the reviewers agree that the idea is well motivated, the paper is well structured and organised, and the experimental results are adequate.